# V1-bypassing suppression leads to direction-specific microsaccade modulation in visual coding and perception

Yujie Wu [1], Tian Wang[1,2], Tingting Zhou[1], Yang Li[1], Yi Yang [1], Weifeng Dai[1], Yange Zhang[1], Chuanliang Han[1] & Dajun Xing [1] ✉

Microsaccades play a critical role in refreshing visual information and have been shown to have direction-specific influences on human perception. However, the neural mechanisms underlying such direction-specific effects remains unknown. Here, we report the emergence of direction-specific microsaccade modulation in the middle layer of V2 but not in V1: responses of V2 neurons after microsaccades moved toward their receptive fields were stronger than those when microsaccades moved away. The decreased responses from V1 to V2, which are correlated with the amplitude of micro-saccades away from receptive fields, suggest topographically location-specific suppression from an oculomotor source. Consistent with directional effects in V2, microsaccades function as a guide for monkeys' behavior in a peripheral detection task; both can be explained by a dynamic neural network. Our findings suggest a V1-bypassing suppressive circuit for direction-specific microsaccade modulation in V2 and its functional influence on visual sensitivity, which highlights the optimal sampling nature of microsaccades.

Saccades in primates are critical for active sampling of visual inputs from complex environments. Even during fixation, eyes often exhibit small (typically <1°) and involuntary movements (microsaccades), which can help the brain locally sample stimuli at a finer level of detail[1]. Similar to large saccades (typically >1°), microsaccades have been shown to have effects on refreshing visual information, such as enhancing visual details[2], restoring fading objects[3,4], and counteracting visual filling-in[5].

Traditionally, microsaccades in all directions have been assumed to make similar contributions to refreshing visual information[3,6,7]. However, recent studies have reported a direction-dependent micro-saccade modulation on human visual perception[8,9], which indicates that information in the visual cortex might be influenced by micro-saccades in a direction-specific manner[10,11]. One possible way that the visual cortex may obtain directional microsaccade modulation is through feedback connections from the high-level cortex. For example, the frontal eye field (FEF) is thought to be crucial for microsaccade

deployment[12] and can provide top-down attentional modulation in visual cortex[13,14]. On the other hand, the tight link between directional modulation and microsaccades in the superior colliculus (SC) intro-duced the possibility of a subcortical pathway[15]. The SC not only controls microsaccade generation[12,16] and other orienting movements[17], but also sends saccadic modulation to the visual cortex through the pulvinar[18]. Both the high-level cortex and subcortical region are possible sources of directional microsaccade modulation in the visual cortex. However, to date, how microsaccades modulate the visual cortex remains unclear; and the previous studies[19–24] have shown little evidence for direction-specific microsaccade modulations in early visual cortex.

In the current study, we observed direction-specific microsaccade modulation in V2 and revealed a suppressive effect by comparing the dynamic neural responses of macaque V1 and V2 after microsaccade onset. We then investigated the circuit mechanism by examining laminar response patterns in V2. Based on anatomy studies, feedback

[1]State Key Laboratory of Cognitive Neuroscience and Learning & IDG/McGovern Institute for Brain Research, Beijing Normal University, Beijing 100875, China. [2]College of Life Sciences, Beijing Normal University, Beijing 100875, China. ✉e-mail: dajun_xing@bnu.edu.cn

modulation targets the superficial layer of the visual cortex[25] while subcortical modulation from the pulvinar targets the middle layer of the extrastriate cortex[26,27]. We found that the directionally suppressive effect on neural activity appeared first in the V2 middle layer, indicating a subcortical pathway. Finally, we examined the behavioral consequences of the suppressive direction-specific microsaccade modulation in a monkey detection task and constructed a computational model to link neural responses in V2 and monkey behaviors.

## Results

We recorded the multi-unit activity (MUA) and local field potential (LFP) in V1 (DQ 6 sessions, DK 9 sessions, DS 8 sessions) and V2 (DQ 17 sessions, DK 8 sessions, DS 7 sessions) of three awake macaque monkeys with a linear multielectrode array (V-probe) (Fig. 1a; cortical

localization and laminar assignment for V1 and V2 are detailed in the methods). The probe was inserted perpendicular to the cortical surface to record responses from all layers within a column of V1 or was inserted into V2 underneath V1 on a daily basis. The receptive fields (RFs) overlapped across channels within each area in each session and overlapped between two areas across sessions (Fig. 1b). During each recording trial, the three monkeys were trained to fixate on a spatial range (radius = 1°) around a small dot (radius = 0.1°) in the center of screen, while a black or white square (4–6°) with uniform luminance was presented (for 3 s) to fully cover the RFs of the recorded sites (Fig. 1a). The eye movement in each trial was recorded by an infrared camera and microsaccades were identified based on the velocity and acceleration of eye movement[28]. A single-trial eye movement trace example is shown in Fig. 1c, and the jump-like periods shown in red

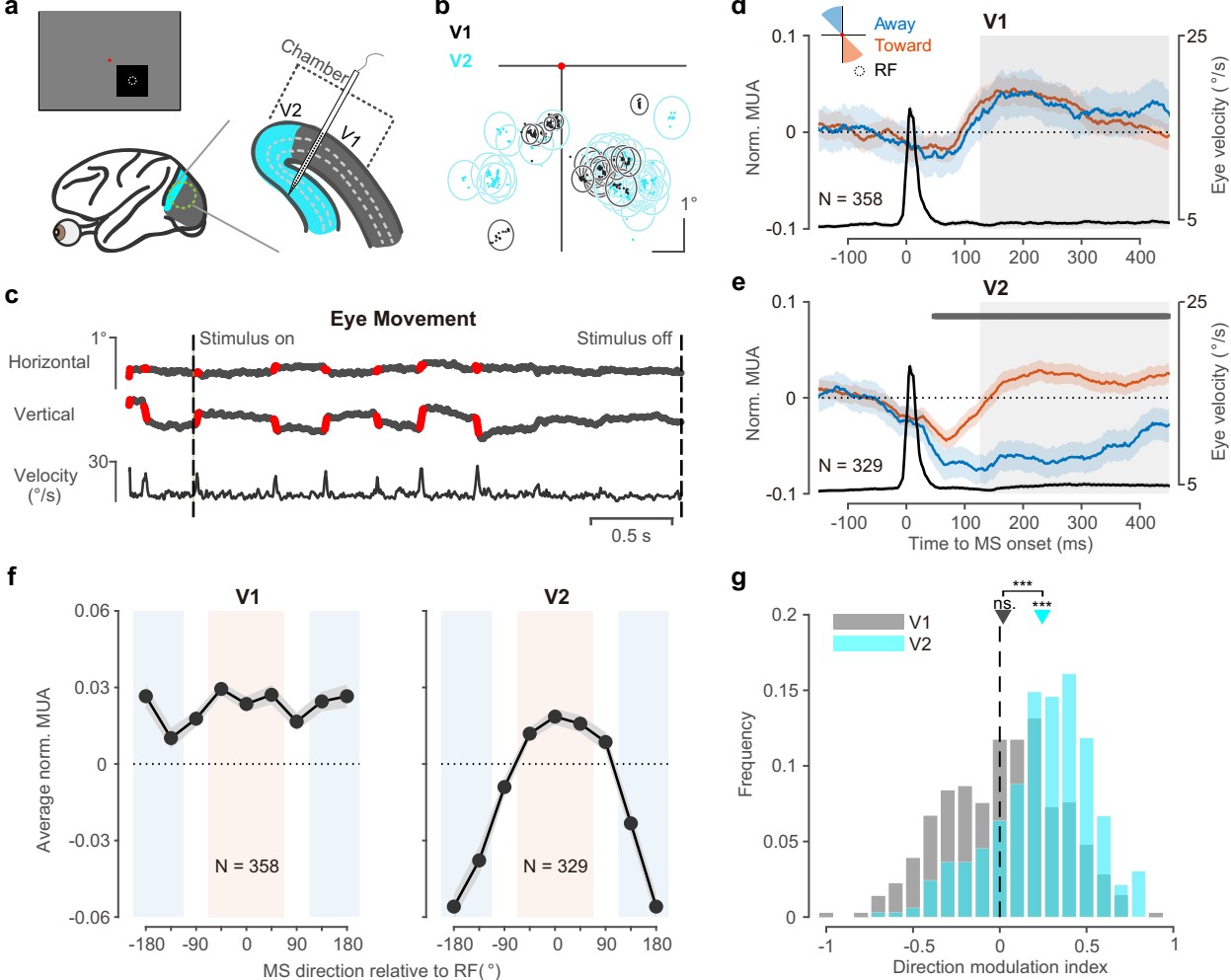

**Fig. 1 | Direction-specific microsaccade modulation in V2 but not in V1. a** A uniform square stimulus with its center located at the RFs of the recording sites was used in the fixation task. The spiking activity and local field potential were recorded with a V-probe (Plexon, 24 channels, interchannel spacing 100 µm). In each recording session, the linear probe was inserted perpendicularly into V1 (gray region) or V2 (cyan region, underneath V1) to record neurons from all layers. **b** Spatial distributions for RF centers (dots) of recorded units from all sessions. The circles represent the mean location (center) and size (radius) of the RFs from each session. **c** The horizontal and vertical eye-movement locations from one example trial were recorded by an infrared camera. The identified microsaccade-generation periods are marked in red. The vertical dashed lines indicate the onset (*t* = 0 s) or offset (*t* = 3 s) of the stimulus. **d** Normalized MUA (mean ± s.e., orange and blue lines in the left vertical axis) in V1 and averaged velocity of eye movement (the black line in the right vertical axis) around microsaccade (MS) onset. Microsaccades were

classified into two groups based on their direction relative to the location of the RF: toward (45° in range, orange) and away from the RF (blue). The response from each site was normalized by the maximum value evoked by the darkest and brightest squares. The black line indicates the averaged velocity of the eye movements aligned by microsaccade onsets. The gray shadow indicates the period (126–450 ms) used for calculating the average response in further analysis (**f, g**). **e** Normalized MUA (mean ± s.e.) and averaged eye velocity in V2 around microsaccade onset. The gray bar at the top indicates the significant time (two-sided *t* test with Bonferroni corrections). **f** Microsaccade direction tuning for the averaged responses (±s.e.) of V1 and V2 from the period shown in (**c**). Red and blue shadow regions indicate two direction bins used for the calculated direction modulation index in (**g**). **g** Distribution of direction modulation in V1 and V2 (two-sided *t* test, \*\*\**p* < 10⁻¹⁰, 'ns.' for V1: *p* = 0.26, V1 vs V2: *p* = 2.9 × 10⁻²⁰, V2 vs 0: *p* = 1.9 × 10⁻⁴⁰). Source data are provided as a Source Data file.

were identified as microsaccades. Consistent with previous studies[7,29], the peak velocity and amplitude of each microsaccade were highly correlated (Pearson's correlation, DQ: $r = 0.90$, $p < 10^{-10}$; DK: $r = 0.86$, $p < 10^{-10}$; DS: $r = 0.91$, $p < 10^{-10}$) in all three monkeys (Supplementary Fig. 1), which indicated the good quality of the microsaccade selection.

**Direction-specific microsaccade modulation in V2 but not in V1**
To test whether microsaccades could modulate neural responses in V1 and V2 in a direction-specific manner, we compared the averaged dynamic responses after microsaccade onset in two directions (Fig. 1d, e): when the microsaccades moved from the fixation point to the RFs (toward direction, marked in orange in Fig. 1a) and when the microsaccades moved from the fixation point to the opposite side of the RFs (away direction, marked in blue in Fig. 1a). The microsaccade-triggered responses (Fig. 1d, e) were defined as the averaged responses lined up with each microsaccade onset ($t = 0$) in a given direction. To avoid the contamination from stimulus-driven dynamics, all microsaccades and neural responses were restricted to the time period from 500 ms after stimulus onset to the end of the visual stimulus. We found strong direction-specific modulations in V2 responses after microsaccade onset but not in V1 responses: V1 responses after microsaccades moved toward and away from the RFs were both biphasic with no significant differences (Fig. 1d). However, V2 responses were much stronger after microsaccades moved toward RFs (Fig. 1e). We found consistent results in V1 and V2 for all three monkeys and the directional effect in V2 was robust across sessions (Supplementary Fig. 2). We further divided the whole visual field into eight direction bins relative to the toward direction (the direction from the fixation point to the RF location) and calculated the averaged MUAs after microsaccade onset (126–450 ms shown as the gray region in Fig. 1d, e) in each direction bin (Fig. 1f). We found that the averaged cortical responses were tuned to microsaccade direction in area V2 (one-way repeated-measures ANOVA, $F(7, 2296) = 55.3$, $MS_e = 0.004$, $p < 10^{-10}$. Post hoc comparisons, 0° vs. −180/−135/−90/135°: $p$s < 0.001, 0° vs. −45/45/90°: $p$s > 0.91, with Bonferroni corrections) but not in area V1 (one-way repeated-measures ANOVA, $F(7, 2499) = 3.68$, $MS_e = 0.004$, $p < 0.001$. Post hoc comparisons, 0° vs. all other directions: $p$s > 0.05, with Bonferroni corrections). The neural responses after microsaccade onset revealed a transition of spatial modulation from a nondirectional pattern in V1 to a directional pattern in V2.

To quantitively evaluate the directional modulation strength of microsaccade, we defined the directional modulation index (DMI) as the ratio of the difference to the summation of the averaged responses in the two bins of interest (Fig. 1f, one bin of 0° ± 67.5°, marked in orange, and the other bin of 180° ± 67.5°, marked in blue). A larger DMI (above 0) indicates that responses under the toward condition are stronger than responses under the away condition, while values near 0 indicate comparable response strength under the two direction conditions (Fig. 1g). The DMIs in V1 (0.02 ± 0.33, mean ± SD) showed no significance with zero (two-sided $t$ test, $t(357) = 1.14$, $p = 0.26$) while the DMIs in V2 (0.25 ± 0.29, mean ± SD) were significantly higher than those in V1 (V2 vs V1: two-sided $t$ test, $t(685) = 9.52$, $p < 10^{-15}$; V2 vs 0: two-sided $t$ test, $t(328) = 15.35$, $p < 10^{-35}$). The distributions of DMI of the V1 and V2 populations were consistent across 5 different levels of the signal-to-noise ratio, which was defined by the mean visual response divided by the standard deviation of baseline activity (Supplementary Fig. 3). All findings (Fig. 1) were replicated after selecting channels at different signal-to-noise levels (Supplementary Fig. 4).

We further validate our findings (the difference in direction-specific microsaccade modulation between V1 and V2) by including the V1 units simultaneously recorded in some V2 sessions ($n = 18$ penetrations). In these sessions, in addition to all layers of V2, deep layers or middle to deep layers of V1 were also recorded by electrodes on a single probe (Supplementary Fig. 5a). With a stimulus fully covering all recorded RFs from both regions, the averaged neuronal dynamics in V1

($n = 104$) after microsaccades toward and away from RFs overlapped with each other, and neuronal activity in V2 ($n = 183$) showed a significant difference between the two directions (Supplementary Fig. 5b).

Such directional modulation relative to the RF location after microsaccade onset was also found in local field potentials (Supplementary Fig. 6). The gamma-band power (50–85 Hz) was higher after microsaccades moved in the toward direction than after microsaccades moved in the away direction (Supplementary Fig. 6a). Moreover, gamma power exhibited clear microsaccade-direction tuning in V2 but not in V1 (Supplementary Fig. 6b). The direction modulation strength of the gamma-band power in V2 was significantly higher than that in V1 (Supplementary Fig. 6c, V2: 0.20 ± 0.26, V1: 0.002 ± 0.29, mean ± SD, two-sided $t$ test, $t(685) = 8.71$, $p < 10^{-15}$) at the population level.

**Directional microsaccade modulation is due to an extraretinal source**
The direction-specific microsaccade modulation (DSMM) in V2 can be explained by either a mechanism from an extraretinal source (microsaccade mechanism) or a mechanism due to RF sensitivity to microsaccade-induced stimulus motion on the retina (RF mechanism). Although we used large squares with uniform luminance to avoid any stimulus changes in the classical RFs during microsaccade generation, the nonclassical RF of V2 might still capture the microsaccade-induced motions/displacements over the edge of the squares (surround modulation[30]).

To rule out the possibility that DSMM was due to the microsaccade-induced square motions relative to the V2 RF, we conducted two control experiments. First, we recorded neural responses from V1 and V2 while monkeys performed a fixation task with a blank screen (there were no square stimuli, and the screen only had a gray background). We then performed an analysis similar to the previous section (Fig. 1d, e) on the dataset. We found that under the blank condition, the V2 response after microsaccade onset was also stronger for the toward condition than for the away condition (Fig. 2a), with no significant difference in V1 between the two conditions. For simultaneous recordings in both V1 and V2, the results under the blank condition were consistent with the results from separate recordings in V1 and V2 (Supplementary Fig. 5c). To further exclude the stimulus influence mediated by firing rates, we analyzed the relationship between the direction modulation index and baseline firing rates before microsaccades. There was no correlation between DMI and baseline activities in V1 ($r = 0.07$, $p = 0.199$) or V2 ($r = 0.11$, $p = 0.051$). We also divided all channels into 5 groups based on quintiles of baseline firing rates (Fig. 2b) and found that direction modulation in V2 was significantly higher than 0 and was stronger than those in V1 at all baseline levels. These results suggest that the response modulation after microsaccade onset in V2 is independent of visual stimuli and the level of response from visual cortex.

Then we recorded neural activity in V2 with the RF covered by a square with small motions that mimicked the trajectories of microsaccades from a monkey (DS). In each trial of the second control experiment, a square with constant luminance (black or white) moved its center location every 0.5 s in a small range (<1.2°) around the RF centers of the recording sites (Fig. 2c). If the direction modulation was due to motion selectivity of nonclassical V2 RFs (neurons whose RF surrounds were activated by the microsaccade-induced edge motion or object motion) after each microsaccade, then the square motion would induce the direction modulation in V2 responses. However, we did not find a significant difference in the responses after sham microsaccade motions (neural responses after square motions) between the toward and away conditions, while the direction modulation of real microsaccades in the same control experiment was still robust (Fig. 2d).

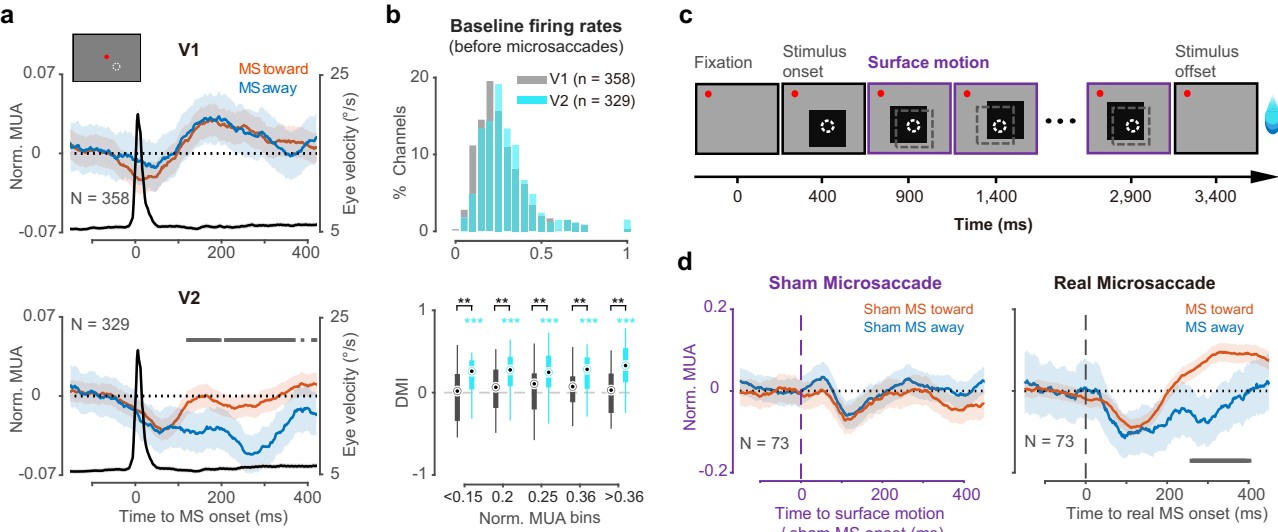

**Fig. 2 | Direction-specific microsaccade modulation in V2 is not stimulus-driven. a** Normalized MUA around microsaccades in V1 and V2 in blank conditions. Black curves indicate the velocity of eye movement (mean ± s.e., right vertical axis). **b** Distribution of normalized baseline firing rates for V1 (*n* = 358 sites) and V2 (*n* = 329 sites) before microsaccades. Box plots show the median (middle points), 25th, 75th percentile (box), and 5th and 95th percentile (whiskers) of the direction modulation index (DMI) among the V1 and V2 populations at each baseline level (defined by quintiles for baseline firing rates). Direction modulation in V2 was significantly stronger than those in V1 at all baseline levels (two-sided *t* test with FDR corrections, $p = 4.35 \times 10^{-5}$, $p = 1.51 \times 10^{-4}$, $p = 6.2 \times 10^{-4}$, $p = 6.2 \times 10^{-4}$, $p = 1.04 \times 10^{-5}$). Asterisks denote

statistical significance (**$p < 0.001$, ***$p < 10^{-5}$). **c** An example of a stimulus sequence in the control experiment. In one trial, after presenting a blank for 0.4 s, a square appeared for 0.5 s. Then, the location of the square was changed within 1.2° every 0.5 s for 5 times. The uniform surface in the RFs remained constant during each jerk of the square. The centrifugal motion of a stimulus mimics the retinal image change after microsaccades move toward the RFs. **d** Normalized MUA (mean ± s.e.) in V2 after surface motion (*N* = 73) and normalized MUA (mean ± s.e.) in V2 after real microsaccade onset. Gray bars indicate significant differences between the two conditions (two-sided *t* test, Bonferroni-corrected). Source data are provided as a Source Data file.

## Direction-dependent modulation of microsaccades is suppressive

The main experiment and control experiments in the previous sections demonstrated our finding of direction-specific modulation of microsaccade in V2 originating from an extraretinal source. A related question is whether DSMM in V2 might be due to facilitation of the responses of a V2 neuron by microsaccades moving in the toward direction or a suppressive modulation of neural responses by microsaccades moving in the away direction (Fig. 3a).

To understand which neural mechanisms led to DSMM (Fig. 3a), we compared microsaccade-triggered dynamic responses between V1 and V2 in eight directions (Fig. 3b). Dynamic responses in V1 and V2 highly overlapped across time after microsaccades moved toward their RFs, which suggested that under the toward condition, V2 responses were mainly driven by V1 responses via a feedforward projection (Fig. 3a). However, the dynamic responses in V1 and V2 gradually separated as the direction of microsaccade relative to RF locations increased. The significantly lower responses in V2 than V1 after microsaccades moved away from their RFs indicated that suppressive modulation was required to explain V2 dynamic responses in addition to the inherited dynamic response from V1. To quantitatively evaluate the suppression of different directions of microsaccades in V2, we used a bootstrap method to sample the V1 and V2 populations and defined a suppression index as the ratio of the averaged response change from V1 to V2 (during the time period of 126–450 ms after microsaccades) divided by the averaged value of the absolute neural responses of V1 and V2 in a given microsaccade direction. We found a clear U-shaped tuning, with the strongest suppression in the direction opposite of microsaccade (1 ± 0, mean ± SD) and no suppression in the direction of microsaccades (Fig. 3c, 0.11 ± 0.11, mean ± SD, percentile method, *p* > 0.05). We also tested the suppression hypothesis using simultaneous recordings of V1 and V2 (Supplementary Fig. 5e and f). The suppression index from V2 to V1 was calculated for each

penetration and for each direction bin and then averaged across penetrations to obtain the directional modulation (Supplementary Fig. 5f). A similar U-shaped pattern (as in Fig. 3c) was observed for the suppression of neuronal activities from the toward direction to the away direction in V2 (Supplementary Fig. 5f). When compared with the V1 responses, the strong decrease in the V2 response in the away direction from both separate recordings and simultaneous recordings suggests that V1-bypassing suppression contributes to DSMM in V2.

The above results and observations indicate that direction-specific microsaccade modulation is extraretinal and suppressive. We speculate that the suppressive modulation is from a motor signal. If this speculation is true, there might be some relationship between suppressive strengths and microsaccade amplitudes. Therefore, we further tested this hypothesis by binning neural responses based on microsaccade amplitudes separately for toward and away conditions and calculating the correlation between microsaccade amplitudes and averaged firing rates (126–450 ms, the same period where we showed direction modulation) in V1 and V2 (Fig. 3d and e). We found a significant correlation (*r* = −0.7; *p* < 0.05) in the away condition but not in the toward condition in V2: larger microsaccades away from RFs were followed by stronger suppression of firing rates (Fig. 3e). In addition, such a correlation in V1 is not significant in either direction condition. The negative correlation between firing rates and microsaccades away from RFs in V2 further indicates that the suppressive nature of direction-specific modulation in V2 is due to a potential motor source.

## Directional microsaccade modulation first appeared in the middle layer of V2

Our next aim was to determine whether the direction-specific suppressive modulation in V2 is due to a subcortical pathway or a feedback circuit. According to anatomical studies, if the DSMM is caused by a top-down signal from higher cortical regions (Fig. 4a), we can expect DSMM to first appear in the top of the superficial layer of V2 and not in

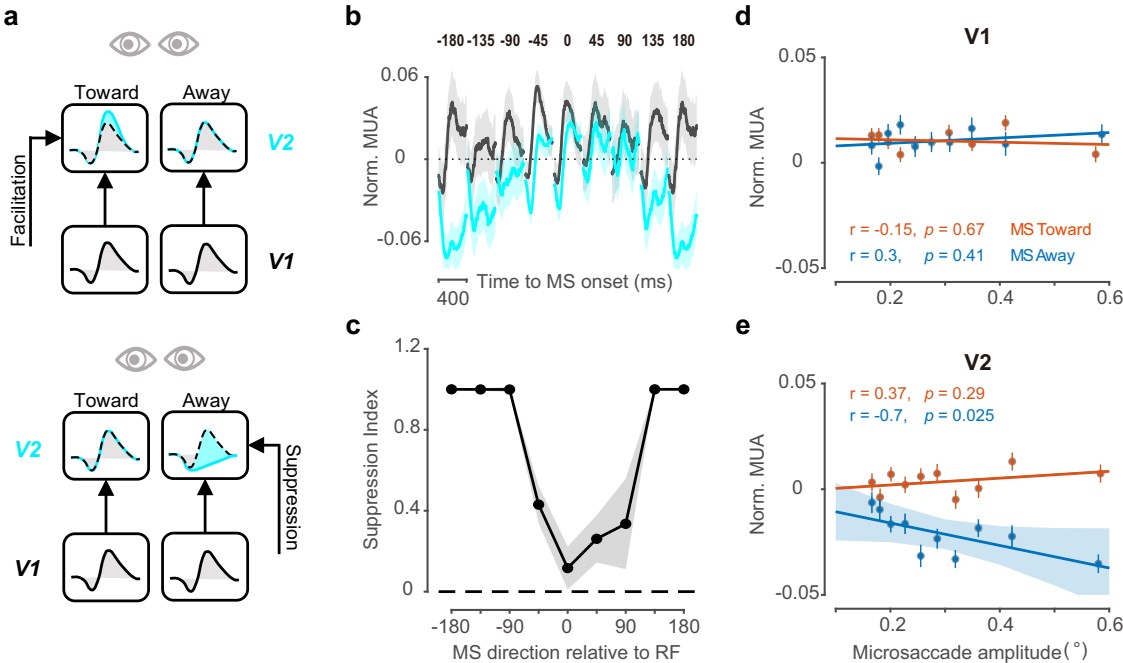

**Fig. 3 | The suppression contributes to direction-specific microsaccade modulation in V2. a** Schematic illustration of two modulation forms. Top, in addition to the feedforward drive from V1, an additional facilitation targets V2 neurons when microsaccades move toward their RFs. Bottom, suppressive modulation targets V2 neurons when microsaccades move away from their RFs. **b** Normalized MUA (mean ± s.e.) in V1 (gray) and in V2 (cyan) after microsaccades with 9 directions relative to RF locations of the recorded sites. **c** The relative direction tuning (MUA mean ± s.e.) of the suppression index on the microsaccade. **d** V1 MUA (averaged and normalized) during 126–450 ms after microsaccades and binned by micro-saccade amplitudes separately in each direction condition. Dots represent group means ($n = 10$ groups). **e** V2 MUA (averaged and normalized) binned by micro-saccade amplitudes in each direction ($n = 10$ groups). Only microsaccades away from RFs showed a significant negative correlation between firing rates and microsaccade amplitudes. Shadow region represents 95% confidence intervals. Source data are provided as a Source Data file.

the middle layer of V2[25]; however, subcortical modulation will cause DSMM to appear first in the middle layer of V2, including layer 4 and layer 3B[26]. Simultaneous recording from each layer in V2 allowed for a detailed investigation of the laminar profile of microsaccade modulation (Fig. 4b).

We assigned each recording site in V1 and V2 with depth information relative to the most superficial layer according to current source density[31,32]. According to the CSD pattern of sink and source, V1 can be divided into 6 layers[33–36]. Because V2 is thinner, all sites can be classified as superficial layer, middle layer, or deep layer. We showed the spatiotemporal laminar dynamics after microsaccade onset for each layer in V1 and V2 (Fig. 4). In contrast to the comparable response pattern in V1 between the toward and away conditions (Fig. 4b, top; Fig. 4c, left), the laminar responses in V2 exhibited strong suppressive modulation for microsaccades moving away from their RFs in all three layers (Fig. 4b, bottom; Fig. 4c, right). The laminar distribution of the DMIs in V1 and V2 from the three monkeys are shown in Fig. 4d. In V1, we did not find significant direction modulation in any layer (two-sided $t$ test, L2/3 $t(101) = 1.95$, $p = 0.054$; L4B $t(34) = 0.66$, $p = 0.514$; L4C$\alpha$ $t(45) = -0.56$, $p = 0.576$; L4C$\beta$ $t(45) = -1.68$, $p = 0.100$; L5 $t(62) = 0.33$, $p = 0.745$; L6 $t(65) = 1.84$, $p = 0.070$). In V2, all layers exhibited significant directional modulation (two-sided $t$ test, L2/3, $t(115) = 9.55$, $p < 10^{-10}$; L4, $t(106) = 10.92$ $p < 10^{-10}$; L5/6, $t(105) = 6.47$, $p < 10^{-8}$, after corrections) and the direction modulation strength was strongest in the middle layer (one-way ANOVA: $F(2,326) = 4.92$, $p = 0.008$; Bonferroni-corrected multiple comparisons: SG vs. G, $p = 0.237$; IG vs. G, $p = 0.005$; SG vs. IG, $p = 0.455$).

In addition to the strongest modulation strength in the V2 middle layer, we can see that the significant difference between the toward and away conditions first appeared in the middle layer at the population level (Fig. 4c). To test the latency difference in the three layers, we estimated the onset of direction modulation in the three V2 layers. We

bootstrapped the penetrations and averaged responses under the toward and away conditions in each layer and used the earliest time with significant direction modulation as the latency. Figure 4e shows the distribution of the bootstrapped population latency of the three layers. The latency in the middle layer (median = 104 ms) was significantly shorter than that in the superficial (median = 126 ms) and deep (median = 226 ms) layers (Kruskal–Wallis test, multiple comparisons with Bonferroni correction, $ps < 10^{-5}$). Taken together, both the modulation strength and modulation latency results supported a V1-bypassing feedforward circuit of direction modulation in V2.

## Behavior detection performance shows directional micro-saccade modulation

Our neurophysiological results indicate that microsaccades will cause a more suppressed V2 cortical state in neural populations with RFs in the opposite direction than in neural populations with RFs in the toward direction. However, what is the functional influence of the neural state related to microsaccade direction modulation is still unknow. We speculated that cortical excitability in the V2 population might further affect the coding of input signals in V2 as well as drive downstream cortical regions related to decisions for perception and behaviors. In this section, we wanted to test whether the DSMM effect also exists in behavior. Specifically, we asked whether the dynamic visual sensitivity of macaque monkeys was also modulated by microsaccades in a direction-specific way.

We trained two monkeys to conduct a near-threshold peripheral target detection task (DN, 19 sessions, 26354 trials; DS, 21 sessions, 23,356 trials). In each trial, after a random delay (0.8–2 s), a bright target was presented for a short period (20–40 ms) on the horizontal meridian on either side with the same possibility (Fig. 5a). Monkeys could earn a reward (water) by maintaining fixation until the central fixation dot vanished and then making a saccade to the target

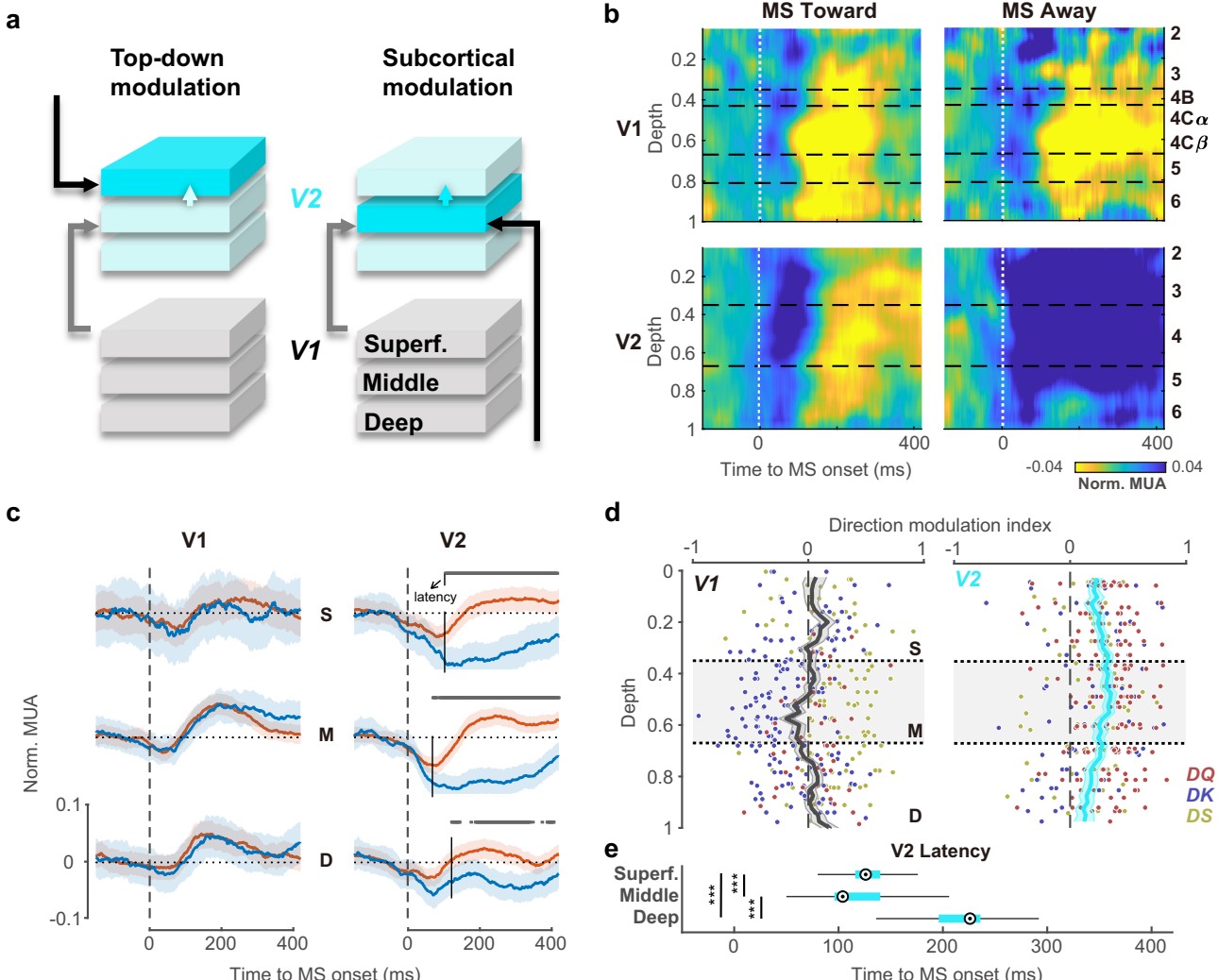

**Fig. 4 | Laminar profile of microsaccade modulation in V1 and V2. a** Schematics of laminar circuits based on two hypotheses. Left panel, top-down modulation targets on the superficial layer in V2, shown by the black arrow. Feedforward connections from V1 superficial layer targets on the middle layer of V2 (the gray arrow) and interlayer feedforward connections from the middle layer to the superficial layer (the gray arrow). If DSMM is from the high-level cortex, the DSMM would first appear in the superficial layer in V2 (marked in dark blue). Right panel, subcortical connections target the middle layer of V2 (the black arrow). If DSMM is from a subcortical region, DSMM would first appear in the middle layer in V2 (marked in dark blue). **b** Laminar pattern of microsaccade dynamics averaged from multiple probe locations in V1 ($N = 23$) and V2 ($N = 32$). MUA response strength was coded by color. **c** Left, normalized MUA (mean ± s.e.) in V1 around microsaccade onset in the superficial layer (top), middle layer (middle) and deep layer (bottom). The toward condition is marked in orange, and the away condition is marked in

blue. Right, same as left, but the normalized MUA is from V2. Vertical dashed lines represent the time of microsaccade onset. The black vertical lines and arrow represent the latency of direction modulation (S: 104 ms, M: 70 ms, D: 122 ms) which was defined by the first time point of significant differences between the toward and away conditions (two-sided paired $t$ test with Bonferroni corrections). **d** Left panel, the laminar distribution of the direction modulation index in V1. The vertical black line indicates the mean (±s.e.) of each depth bin. Each dot represents one site in V1 ($N = 358$, red: sites from monkey DQ, yellow: DS, blue: DK). Right panel, laminar distribution of the direction modulation index in V2 ($N = 329$). **e** Box plots show the median (middle points), 25th, 75th percentile (box) and 5th and 95th percentile (whiskers) of population latency for each layer in V2 ($n = 32$ penetrations) after bootstrapping ($n = 1000$, post hoc test of Kruskal–Wallis test with Bonferroni correction, ***$p = 2.87 \times 10^{-9}$). Source data are provided as a Source Data file.

location. To measure the change in visual sensitivity, we selected a luminance at which each monkey had a detection rate accuracy of 75% (Fig. 5b).

We calculated the detection accuracy for targets appearing up to 0.9 s after microsaccades in bins of 40 ms and ensured that no other microsaccades occurred between the microsaccade of interest and the target. From the temporal dynamics of detection accuracy (Fig. 5c, d), surprisingly, we found that after microsaccades moved toward the target direction, the detection performance was significantly better than that after microsaccades away from target locations (during 0.3-0.42 s for DN and 0.58-0.7 s for DS, $p < 0.01$, bootstrap percentile test after Bonferroni corrections for multiple comparisons across time). Averaged accuracy after microsaccades

(300–700 ms) was higher under toward condition in both monkey DN and DS (mean ± SD, DN toward: 0.72 ± 0.01, DN away 0.70 ± 0.01, $p = 0.035$; DS toward: 0.74 ± 0.02, DS away 0.69 ± 0.02, $p = 10^{-5}$, Fig. 5c, d). This suggests that microsaccade direction could modulate visual sensitivity on behavior and the modulation trend was consistent with what we found for neural activities in V2 (both MUA and gamma power of LFP).

In addition to the direction effect after microsaccades, interestingly, during microsaccade generation (−20-20 ms), the detection accuracy of both monkeys was also modulated by direction: the target that appeared in the opposite direction of ongoing microsaccades was harder to detect ($p < 0.05$). This modulation during microsaccade generation is similar to what has been found in SC neurons[15].

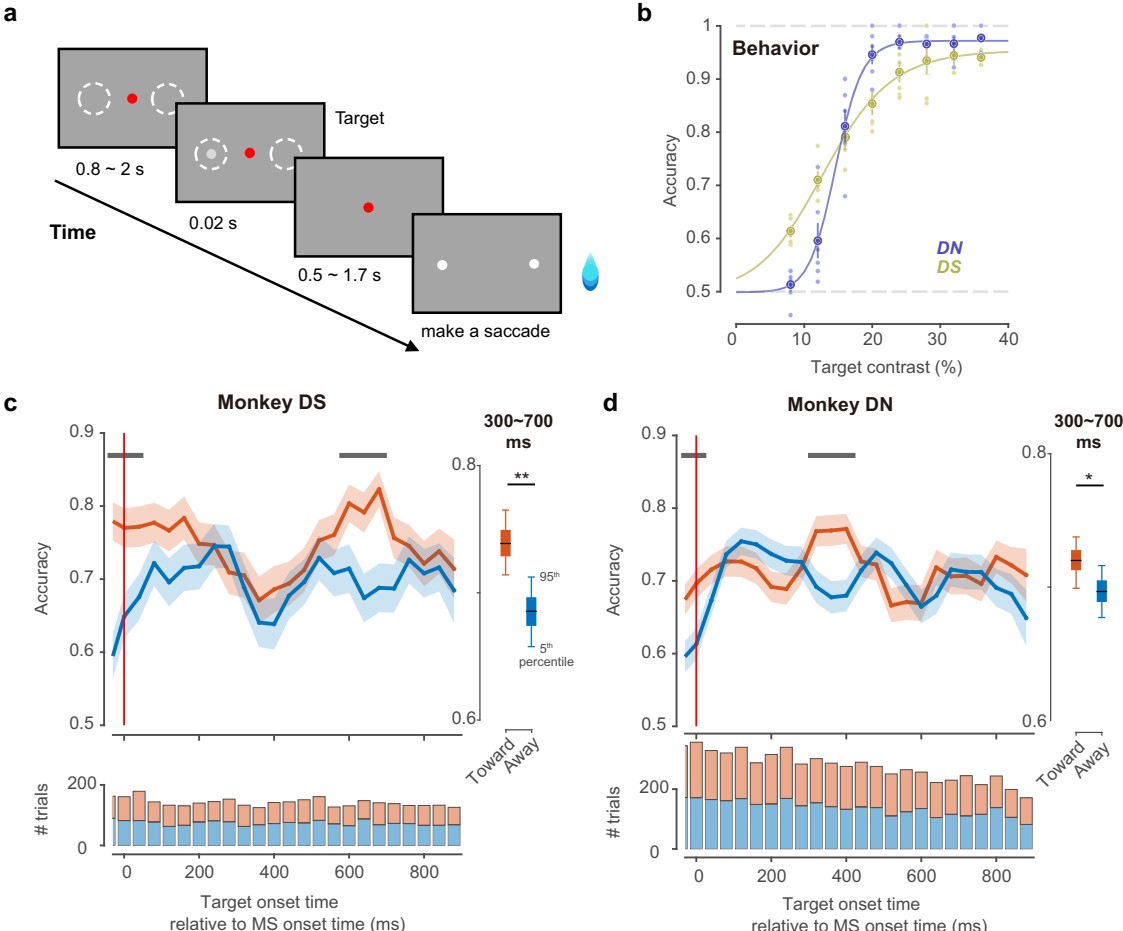

**Fig. 5 | Direction modulation of microsaccades in behavioral detection tasks.**
**a** Trial procedure in a two alternative force choice (2AFC) detection task. Monkeys fixated on a central red dot (0.2°) for at least 0.3 s to start a trial. After a variable blank period, a small (0.35°) target appeared for only 0.02 s on the right side or left side (eccentricity = 5°). White dashed circles indicate the potential target locations in a block, which is not visible to monkeys. Monkeys had to hold fixation until the end of a trial. Once the fixation point disappeared, monkeys could earn a reward by making a saccade to the target location (left white dot in the example trial). All graphic elements are not precisely scaled or luminant. In particular, the target was shown to be much smaller and weaker in luminance in the experiment. **b** Averaged detection accuracy (mean ± s.d.) of monkey DS and DN from threshold-testing sessions ($n = 6$). We selected a target contrast of ~16% (target luminance = 53.13 cd/m², background = 45.8 cd/m²) in formal blocks

which corresponds to a threshold near 75% accuracy. **c** Top, change in detection accuracy (mean ± s.d.) from monkey DN as a function of target onset time relative to a microsaccade. We used a sliding window of 120 ms (3 bins). The red vertical line indicates the onset time of microsaccades. Gray bars indicate significant differences between the two conditions (one-sided percentile method of bootstrap test, $p < 0.01$ with Bonferroni corrections). The box plot shows the median (middle points), 25th, 75th percentile (box) and 5th and 95th percentile (whiskers) of the averaged accuracy during a period of 300–700 ms from 1000 bootstrap samples (one-sided percentile method of bootstrap test, **$p = 10^{-5}$). Bottom, the number of trials used per time bin at the top. **d** Same as (**c**) but for monkey DS (one-sided percentile method of bootstrap test, *$p = 0.035$). Source data are provided as a Source Data file.

## A dynamic model links the neurophysiological findings and behavior

We demonstrated that direction-specific microsaccade modulation occurred in V2 responses and revealed its functional influence on monkey behavioral performance in a visual sensitivity task. In this section, we tested whether the direction-specific suppression we found in V2 and the biphasic dynamic in V1 would be sufficient for interpreting the directional effect on behavioral dynamics in the 2AFC task. Based on our assumption, the spatial modulation of V2 cortical excitability after microsaccades might bring unbalanced strength when driving downstream cortical regions related to decisions for perception and behaviors (e.g., lateral intraparietal area[37]). Motivated by this hypothesis, we designed a hierarchical dynamic neural network with two feedforward visual stages (corresponding to V1 and V2) and one recurrent decision stage inspired by the simplified decision model that was well defined and used to explain the reaction time in the 2AFC task[38]. We first validated our model structure in the current detection

task by replicating the contrast-tuning curve measured psychophysically. Then, we added a microsaccade-related biphasic modulation signal to stage 1 and a directional suppressive signal to stage 2 based on our experimental findings to investigate their contribution to the directional microsaccade effect in behavior dynamics.

In the visual pathway of the computational model (Fig. 6a, red and blue components), the first stage ($P$) contained two components that represent different populations in V1 that received visual input in either the left ($P_L$) or right ($P_R$) visual field. The second stage ($V$) contained two components, representing two local populations of neurons in V2, were directly driven by the corresponding components at the first stage. The last stage ($D$) in the model represented a decision area for determining which side of the visual field had a target based on sensory information at stage 2. Due to the competing nature of the two alternative choices in the 2AFC task, the two components in the third stage mutually inhibit each other. In each trial, the decision for the left or right side was based on the activity of

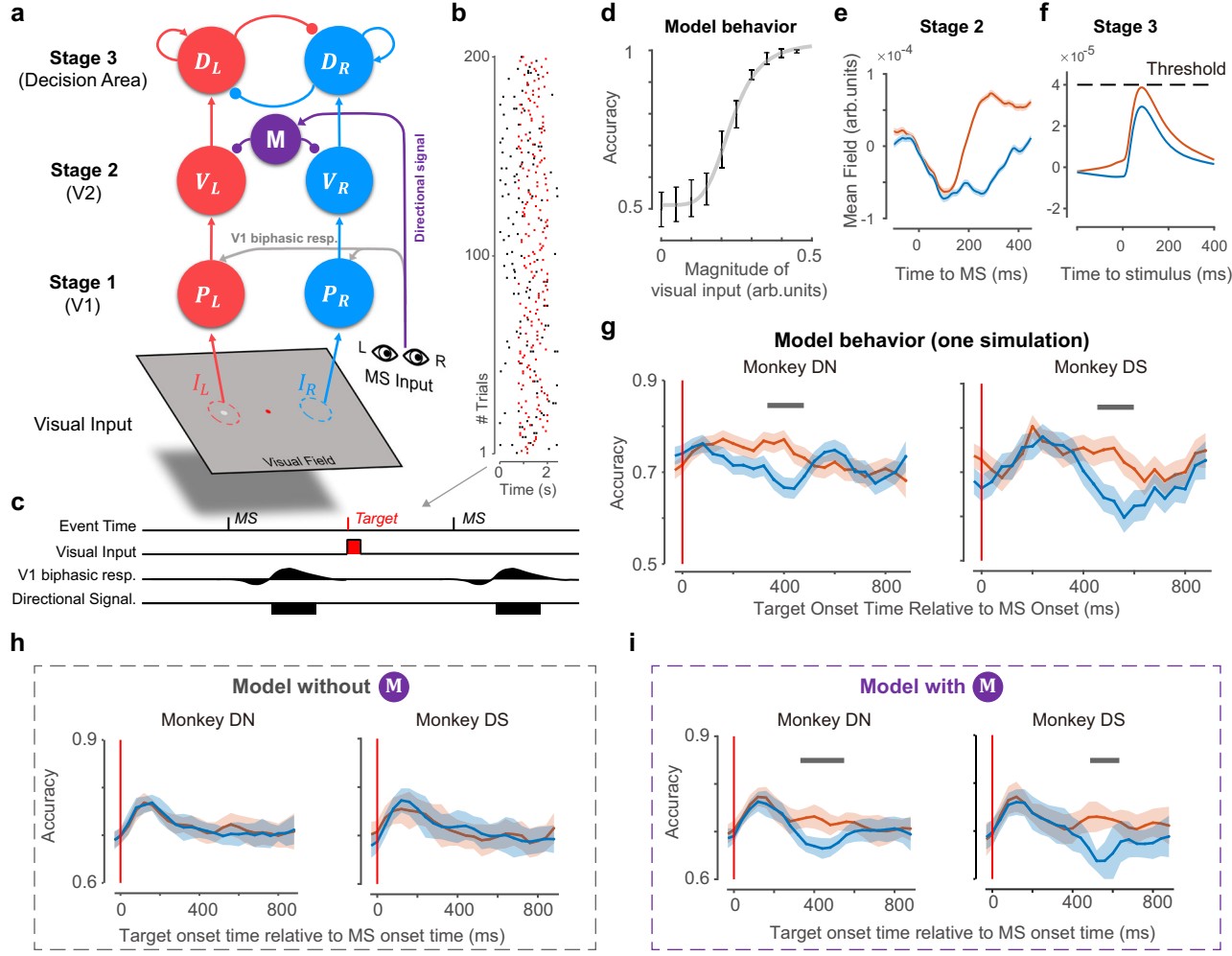

**Fig. 6 | A dynamic model links neurophysiology and behavior. a** Structure illustration of the dynamic recurrent network composed of two visual stages (*P* and *V*), a mutually inhibited decision stage, and two modulation components (biphasic kernel and *M*). The first stage (*P*) was stimulated by a 20-ms current when there was a visual target in the RF. Each microsaccade triggered a biphasic modulation on both components in stage 1 (*P*) and triggered a directional signal to *M*. *M* only suppresses the second stage (*V*) with RF in the opposite direction of microsaccades. **b** The onset time of microsaccades (black dots) and targets (red dots) in example trials from the 2AFC detection task. Targets appeared randomly within 0.8-2 s in each trial. **c** Model input from a time period in one example trial, including visual input to layer 1, biphasic input to stage 1, and directional signal input to *M*. **d** Mean accuracy (±s.d.) across simulations (*n* = 30, each simulation was composed of one repetition of all possible target locations and target onset times) for each level of visual input strength in arbitrary units (arb. units). **e** Model response (mean ± s.d.) in stage 2 (*V*) around the microsaccade, which showed similar direction modulation to

the real data in Fig. 1. **f** Model response (mean ± s.d.) from *D* around target onset across trials in which targets appeared 0.16-0.4 s after the previous microsaccades. **g** Simulated behavior accuracy (mean ± s.d.) as a function of microsaccade-target interval by using microsaccade statistics and target location information from the real experimental data of monkey DN (left panel) or monkey DS (right panel). Gray bars indicate period with significant differences between the two conditions (one-sided percentile method of bootstrap test with Bonferroni corrections, *p* < 0.05). **h** Simulated behavior accuracy (mean ± s.d.) of the control model after removing component *M* across 15 simulations (all trials from all empirical sessions from one monkey compose one simulation). **i** Simulated behavior accuracy (mean ± s.d.) of the full model across 15 simulations. Gray bars indicate period with significant differences between the two conditions (one-sided percentile method of bootstrap test with Bonferroni corrections, *p* < 0.05). Source data are provided as a Source Data file.

model components at the third stage: the component whose activity reached the threshold for the longest time gave a left or right decision. When changing the magnitude of visual input, the accuracy curve of hierarchical dynamic model could be well fitted by a Naka-Rushton function (Fig. 6d) which shows a consistent pattern with monkeys' behavior (Fig. 5b) and indicates good validity of the model structure for the 2AFC detection task.

Next, we chose the input magnitude with a 75% correct rate as we did on monkeys and added microsaccade-related components to the hierarchical model (Fig. 6a, gray and purple components). Both components at stage 1 received biphasic modulation from our experimental data of V1 while the two components at stage 2 received direction-dependent suppressive modulation (*M*), which took an input triggered by each microsaccade. Finally, the modulated sensory

information at stage 2 is transmitted to the two populations in the decision area (*D*).

We simulated the V2 response (from stage 2) and behavior results (based on stage 3) trial by trial with the model using real target and microsaccade information (including onset time and location/direction) from monkeys DS and DN in their behavior sessions (Fig. 6b). After averaging the mean field activity from stage 2 by aligning to the time of microsaccade onset, the direction modulation observed in V2 was simulated successfully (Fig. 6e): average activity is stronger after microsaccade toward the population RF. Microsaccades away from the target location suppress the target response at stage 3 (decision signal), preventing the population with RF on the target side from reaching the decision threshold (Fig. 6f). More importantly, after averaging the accuracy across all simulation trials and performing the

same bootstrapping analysis as we did on monkey data, the behavior modulation pattern of two monkeys was also simulated: after microsaccade onset, the accuracy was significantly higher under the toward condition than that under the away condition (Fig. 6g). To further investigate the contribution of the modulation signal to behavior dynamics, we compared the simulation results (15 simulations) of the full model with those of a control model without a directional component (M). After averaging the dynamics from all 15 simulations, the performance of the control model exhibited decreased accuracy during microsaccade generation and no direction difference was found (Fig. 6h). In contrast, when a directional microsaccade-related signal was projected to stage 2 in the model, directional modulation appeared after microsaccades (Fig. 6i). The comparison between the control model and full model indicates that the directional suppression projecting to V2 may contribute to the directional modulation of behavior dynamics after microsaccades.

When changing the latency of the input current of the direction signal, the average accuracy across time and the accuracy difference between the toward and away conditions remained constant (Supplementary Fig. 7a, b). However, the occurrence time of significant direction modulation changed monotonically (Supplementary Fig. 7c). The link between the time of directional signal and the behavioral effect not only suggests the influence of directional signal on behavior but also provides a potential explanation for individual difference in the two monkeys: delayed directional motor signal to V2 might lead to a delayed modulation effect on visual sensitivity.

Above all, a dynamic neural network model with suppressive directional modulation after microsaccade onset can simulate both V2 responses and behavior dynamics after microsaccades. Furthermore, the model uncovered a possible link between the occurrence of behavior modulation and the latency of the suppressive current to V2.

## Discussion

In the current study, we demonstrated the neural circuit mechanism of direction-dependent microsaccade modulation in macaque V2 and uncovered its functional influence on detection performance in peripheral vision. The comparison of microsaccade modulation in V1 and V2 suggests a suppressive mechanism underlying the directional effect. The negative correlation between firing rates in V2 and microsaccade amplitudes in the away direction indicates an oculomotor source through the subcortical circuit for the suppression, which was supported by the laminar analysis of saccade modulations in V1 and V2. The dynamic neural network simulating V1, V2 and the cortical regions for decision-making in macaques confirms the contribution of the location-specific suppression in V2 on explaining monkey behaviors, which further helps us link our neurophysiological findings to behavioral decision-making.

To the best of our knowledge, the localization and neural mechanisms for the direction modulation of microsaccades in V2 have not been reported before. Most of monkey V2 are underneath the V1 area (see Fig. 1a for demonstration), which makes it harder for recordings. Therefore, only a few studies[23,24,39] have investigated microsaccade modulation in V2, but none have reported and focused on the relationship between RF location and microsaccade direction. Without considering directional modulation in V2, previous studies[23,24,39] have reported response enhancement to visual stimuli with spatial patterns in V2 following microsaccade onset, which is different from our observation of directional suppression in V2. We think this finding is mainly due to different methodologies for stimulus presentation. The displacement of a pattern stimulus in RFs due to microsaccades makes it difficult to investigate the direction modulation from the extraretina source, which is also the case for studies in the middle temporal (MT) region[40,41]. In the current study, we used large squares (4°–6°) with uniform luminance to guarantee that the RFs of V2 (and V1) were fully covered by the surface of visual stimuli. In

this way, V2 RFs would not be directly driven by the edge, or any pattern, of a stimulus even with the consideration of RF shifts caused by microsaccades (<1°). Such an experimental design made the response dynamic after the microsaccade purely reflect a motor signal, without changing the visual stimulus within the RFs of V1 and V2. This method may give us a better chance to see the directional suppression of microsaccades in V2.

Compared to very few studies in V2, there have been more studies on response modulations by microsaccades in primate V1[7,20,21,23,42–46]. Most studies on V1 reported a biphasic response after microsaccades with no evidence for direction specificity to RFs. Our study confirmed the results in macaque V1 (Fig. 1). The strong biphasic modulation we observed in V1 indicates a good microsaccade response, and the laminar pattern of biphasic modulation in V1 was highly consistent with a previous study[46]: the suppression first appeared in the granular and infragranular layers of V1, layers 4 and 6 specifically (Fig. 4b). Compared to the strong nondirectional biphasic modulation, the directional effect was very weak in all V1 layers even when RF eccentricity was controlled using units with overlapping RFs in V1 and V2 (Supplementary Figs. 8 and 9). The possibility of contamination from surround suppression was ruled out by a control experiment with a smaller stimulus (Supplementary Fig. 10). Therefore, V1 might not be a major target of directional modulation by microsaccades. V1 might receive some feedback modulation from the deep layer of V2 as we identified a very weak trend of direction modulation of the microsaccade in the superficial layer of V1. However, the directional modulation in the deep layer of V2 was weakest, which indicates that the feedback modulation would be weak and easy to miss.

The direction-specific microsaccade modulation we found in V2 indicates the importance of the relative direction of microsaccade to RFs for determining their effect on neural activity, which is indispensable when explaining the behavioral benefit of microsaccades toward a visual target. Recently, one study found that the amplitude of the traveling wave in V4 was larger after a large saccade toward RF direction[10]. A more recent study of microsaccades found that only microsaccades toward RFs were followed by attentional enhancement on firing rates of V4 neurons[11]. These two studies, together with our current findings, provide a new perspective when taking microsaccade direction into consideration.

The direction-specific modulation of microsaccades relative to RFs in V2 reflects a consistent topographic map between the oculomotor system and the visual system, which may be important for visual stability which requires knowledge of eye movements. The brain needs to know the direction of the eye movement to avoid misinterpretation of moving environment during microsaccades[47]. And previous anatomical studies found that communication from the oculomotor system to V2 relies on topographic projections[48–50], which might be the neural basis for direction information in the visual cortex. The directional suppression in V2 and in the downstream area could be used for computing and inferring whether the information updating in the brain was due to a rightward microsaccade or leftward move in visual space. The hypothesis was consistent with the results in our control experiment of the moving surface (Fig. 2d), which showed that V2 neurons could reveal the difference between the object moving and the microsaccades.

In addition to the computational importance of visual stability, the directional modulation may help the visual cortex reset excitability. The comparison of dynamic responses between V1 and V2 helped us further uncover the suppressive nature of DSMM. In V2, we found profound and strongest suppression, as well as poor visual sensitivity in behaviors in the opposite direction of microsaccades. Our results imply that microsaccadic suppression can be spatially specific. The continuous changes in suppression strength in the cortical space of V2 after microsaccades result in unbalanced cortical excitability. For a subpopulation whose RFs are in the opposite direction of

microsaccades, the feedforward visual stream from V1 needs to be stronger to counterbalance the suppression, while the visual stream to another subpopulation with RFs in the direction of microsaccades more easily reached the threshold. We found that gamma-band activity, which represents the processing of feedforward visual information in the visual cortex[51], was also modulated by microsaccade direction. The unbalanced strength when driving downstream cortical regions leads to a reset of spatial visual sensitivity after each microsaccade. The asymmetry of cortical excitability and perception in the visual field reflect a sampling outcome after each microsaccade, which is reminiscent of attentional sampling[52–54].

The result of a negative correlation between microsaccade amplitude and neural activity in V2, for microsaccades away from RFs, suggests an oculomotor source of the suppressive mechanism. Neurons in SC can exhibit strong selectivity both for the microsaccade direction and amplitude[16]. In the visual cortex, neurons in area MT were found to receive modulation of saccadic suppression from SC conveyed by pulvinar[18]. The strongest and earliest direction modulation from microsaccades in the V2 middle layer also suggests that such a motor signal might go through a subcortical circuit (see below) instead of a top-down feedback connection as found for attention modulation[55,56]. The suppressive mechanism in the opposite direction of the microsaccade is reminiscent of attention because attention not only enhances but also suppresses cortical responses of irrelevant information[57,58], and the direction of microsaccades can be biased to the attended location[59,60]. Studies have long debated whether the mechanisms of saccades and attention can be dissociated[61]. Our results indicate that the directional modulation by microsaccades in V2 might be dominated by motor signals but not solely by spatial attention. We admit that except modulation from a motor signal, there might be some weak attentional modulation as even in the absence of a stimulus, the monkey may have a bias toward attending a spatial position where the stimulus has been previously shown multiple times. An experiment that controlled both attention and microsaccade could further investigate the contribution and interaction of the two in the visual cortex.

A recent study[11] suggested that attentional enhancement in V4 and IT is triggered only by microsaccades moving toward the attended location (in the RF) but not by microsaccades moving away from the attended location. This result suggests separate effects from attentional modulation and saccadic modulation in the visual cortex; otherwise, there should not be an interaction between the two mechanisms. Our findings may reveal a partial interaction mechanism between microsaccades and attention: the suppression of microsaccades moving away from the attended location might counteract the facilitation effect of attention, whereas a consistent direction for microsaccades and attention might make attentional effects more explicit. Since microsaccades and attention might modulate neural responses in V4 through two separate pathways, future laminar studies in V4 may help to test this hypothesis and further elucidate the influence of microsaccades and attention.

What is the neural circuit behind the direction modulation of microsaccades found in V2? One possible circuit for direction modulation of microsaccades is the pulvinar pathway from the SC to the extrastriate visual cortex, which has long been considered a second visual pathway to the cortex that bypasses the primary geniculostriate path. The SC was found to play a causal role in microsaccade generation[16] and shows strong microsaccadic suppression[62], which is the inhibition of activity at the time of the microsaccade. More importantly, visual neurons in the SC exhibit directionally specific modulation of visual responses to peripheral stimuli both during microsaccade preparation and after microsaccades[15]. Therefore, the SC may be the source of the microsaccade direction modulation signal after microsaccade generation.

Although a direct retrograde transsynaptic tracing experiment showed no two-synapsed projections from SC to V2[63], an indirect tracing experiment found predominant projections from SC to inferior pulvinar (PI) nuclei that project to V2 and V4[64]. Laminar studies found projections from the inferior and lateral pulvinar to the middle layer of V2, specifically layers 4 and 3B[26,49], which is consistent with the laminar profile of the first appearance of DSMM in the middle layer. In addition, in the path from the SC to the extrastriate visual cortex, the medial PI (PIm) and the posterior PI (PIp) were found to have the heaviest projections to MT, while a few projections to V2 and no projections to V1 were found[48]. A subpopulation (40–50%) of PI neurons, including PIm and PIp, conveyed saccadic suppression from the SC to the MT, but these neurons did not convey attentional enhancement modulation[18]. Therefore, we speculated that the pulvinar might serve as a relay station for the suppressive modulation signal of microsaccade direction from the SC to V2. It would be interesting and necessary to examine the directionality of suppressive modulation after microsaccades in the pulvinar.

Another innovative finding is the visual sensitivity in the peripheral was modulated in a spatially specific manner after microsaccades, which is consistent with the modulation of the firing rates in V2. Previous human behavior studies have shown controversial results regarding whether the direction of microsaccades influences visual perception. In an adaptation task, microsaccades in any direction can help restore orientation information in the peripheral visual field[3]. However, microsaccades can influence the visual system in a temporally dynamic way[9]. During the microsaccade preparation period, orientation information at the endpoints of microsaccades in the fovea can be improved, whereas orientation information on the opposite side of the microsaccade can be impaired[8]. Here, in monkeys, we found a similar directional effect during microsaccade generation: there was a substantial decrease in detection performance in the direction opposite of the microsaccades (Fig. 5). Previous behavioral studies have yielded contradictory results on whether there is perceptual suppression during microsaccades, with some reporting an increase in visual threshold[40,65] and others reporting no change[66]. Our result of the directionality of microsaccadic suppression during microsaccade generation illustrates that the relationship between the direction of microsaccades and the target location is essential when evaluating microsaccadic suppression in visual detection tasks. Although the directionality during microsaccade preparation on behavior cannot be explained by our dynamic model, the behavioral result is consistent with directional modulation on neural responses found in SC[15], which implies a combination of both cortical influence and direct subcortical influence on behavior.

We found direction modulation not only during microsaccade generation but also after microsaccade offset. The difference in performance 300–700 ms after microsaccade onset in two directions showed that visual sensitivity was biased toward the microsaccade directions. The sensitivity asymmetry in the two directions relative to the microsaccade direction can be well explained by our neurophysiological results through a dynamic model. Both the behavioral and neurophysiological results indicate that the cortical state was tuned and exhibited excitability fluctuations of the visual system after microsaccades. Moreover, the behavioral results revealed the functional role of direction modulation by microsaccades, which supports the idea of an optimal sampling strategy[6]. Discrete sampling by microsaccades is not only optimal in time, but also optimal in space. Selective sampling may be beneficial for efficient coding and transmission of visual information, as well as for automatically reducing noise without requiring attentional resources.

## Methods
### Preparation of awake monkeys
Four male adult rhesus monkeys (DS, DQ, DN, and DK, Macaca mulatta, 5–7 years old, 4–8 kg) were used. All procedures were conducted in compliance with the National Institutes of Health Guide for the care

and use of laboratory animals, and were approved by the Institutional Animal Care and Use Committee of Beijing Normal University. Under general anesthesia induced with ketamine (10 mg/kg) and maintained with isoflurane (1.5–2.0%), a titanium post was attached to the skull with bone screws to immobilize the animal's head during behavioral training. After the animal had been trained with a simple fixation task (DQ and DK) or had been trained with a detection task (DS), a circular titanium chamber (20 mm in diameter) with a removable lid was fixed over the craniotomy (15 mm anterior to the occipital ridge and 14 mm lateral from the midline) with dental cement for chronic recordings from V1 and V2. Antibiotics and analgesics were used after the surgery.

## Behavioral task

In the fixation task, a trial began when a monkey began fixating on a 0.1° fixation point (FP) presented on a CRT screen. In each trial, the FP was displayed in the center of the screen. The animal's eye positions were sampled at 120 Hz using an infrared tracking system (ISCAN). Within 300 ms of FP presentation, the animal was required to fixate within an invisible circular window (1° in radius) around the FP. Then a trial was initialized. After the monkey maintained fixation on a blank screen for 400 ms, the stimulus was displayed for 3 s, followed by a blank interval of 300 ms. The FP then disappeared, and the animal received a drop of water as a reward. A trial was aborted if the animal's fixation moved outside the fixation window.

In the control task of surface motion, after a trial was initialized by holding fixation within the invisible circular window (1° in radius) for 300 ms, there was a blank period of 400 ms. Then a square (constant luminance of −0.9/0.9) showed up with its location randomly selected from the center locations of 3 × 3 grids (0.3° in width for each grid, the center grid located on the RFs of recording sites). After 500 ms, the square changed its location within the 3 × 3 grids and this procedure was repeated for another four times. Finally, there was a 400-ms blank and then FP disappeared. The animal only received a drop of water as a reward for maintaining fixation until the end of the trial.

In the behavioral detection task, a trial began when a monkey began fixating on a 0.2° FP presented in the center of a CRT screen, with the animal's eye positions were sampled at 120 Hz by an infrared tracking system (ISCAN). During the 2.5 s trial, the monkey needed to hold fixation within a 2° (in diameter) fixation window. After a variable fixation period (0.5-1.3 s or 0.8-2 s), a small (0.35° in diameter) bright dot target appeared in the left or right hemisphere (5° eccentricity) for 20-40 ms. The luminance of the dot changed trial by trial in the threshold-testing sessions (luminance contrast: 8%, 12%, 16%, 20%, 24%, 28%, 32%, 36%) while holding constant (75% threshold/accuracy) in formal sessions. After the dot target disappeared, the monkey still needed to hold the fixation until the end of the trial. After the fixation points disappeared, two white choice dots were presented in each hemisphere, which were potential locations of the target. Monkeys needed to make saccades to the target location previously shown in the trial and hold fixation for 300 ms on the choice dot. Correct saccades were followed by a drop of water as a reward, while incorrect saccades or fixation break were not followed by any reward.

## Electrophysiological recording

We simultaneously recorded neuronal activity from different layers in V1 or V2 using a linear array (V-probe, Plexon; 24 recording channels spaced 100 μm apart, each 15 μm in diameter). The linear array was controlled by a microelectrode drive (NAN Instruments), and the depth of each probe placement was adjusted to extend through all the V1/V2 layers. The raw data were acquired with a 128-channel system (Blackrock Microsystems). The raw data were high-pass filtered (seventh-order Butterworth with 1000 Hz corner frequency), and multiunit spiking activities (MUAs) were detected by applying a voltage threshold with a signal-to-noise ratio of 5.5. The raw data were also low-pass filtered (seventh-order Butterworth with 300 Hz corner frequency) to obtain LFPs. Both MUAs, and LFPs were down-sampled to 500 Hz.

## Visual stimulation

Visual stimuli were generated with a stimulus generator (ViSaGe; Cambridge Research Systems) under the control of a PC running a custom C11 program developed in our laboratory. The stimuli were displayed on a 22-inch CRT monitor (Dell, P1230, 1200 × 900 pixels, mean luminance 45.8 cd/m², 100 Hz refresh rate). The viewing distance was 114 cm. Three types of stimuli were used for fixation tasks. Sparse noise was used to simultaneously map the receptive fields (RFs). Random orientation presentation was used to measure orientation dynamics, align laminar positions, and check the verticality of the probe. Square presentation was used to measure microsaccade modulation (4° in V1, 4–6° in V2). In formal experiment, the squares with eight contrasts (−0.9, −0.65, −0.4, −0.15, 0.15, 0.4, 0.65, 0.9) were used. For monkey DQ and DK, a square stimulus was located at the center of RFs. For monkey DS, two square stimuli (4° in V1, 4–6° in V2) with equal contrast were present at the same time, with one located at the center of the RFs, and the other one located at the symmetric location relative to the fixation point. In the control task of surface motion, the squares with two contrasts (−0.9, 0.9) were used. For each presentation of a square, the location was randomly chosen from the center locations of 3 × 3 grids (0.3° in width for each grid, the center grid located on the RFs of recording sites).

## RF mapping

After manually mapping the RFs of the recording channels, we used sparse noise to identify the precise RF center. The sparse noise consisted of a sequence of randomly positioned (usually on a 13 × 13 or 11 × 11 sample grid) dark and bright squares (0.1°–0.3°, contrast 0.9) against a gray background (luminance 45.8 cd/m²). Each sparse noise image appeared for 20 ms and with at least 50 repetitions. The sequence was cut into small segments based on the trial length. We obtained a two-dimensional map of each channel. The averaged response map of each channel was fitted with a two-dimensional Gaussian function to estimate the center position and radius of each RF (2σ of Gaussian function). The RFs were located within 5° of the fovea.

## Saccade detection

For monkeys DQ, DK, and DN, microsaccades were detected by finding instantaneous eye velocity exceeding 10.4 deg/s, with the start and end points of microsaccades defined as the time when eye acceleration first exceeded 2083 deg/s². For monkey DS, microsaccades were detected by finding eye velocities that exceeded 2 standard deviations in each trial, with the start and end points of microsaccades defined as the time when the eye acceleration first exceeded 1250 deg/s². To avoid noise, we selected microsaccades with amplitudes within 0.1-1°. We only used microsaccades generated after 500 ms of stimulus onset and 600 ms before stimulus offset.

## Normalization

All data analyses were performed using custom programming in MATLAB (MathWorks). Since we used sustained activities after stimulus onset, which were at different levels for various luminance conditions, we normalized the data around microsaccades by the peak response within 40–400 ms after stimulus onset after averaging the responses from the brightest and darkest luminance conditions, and then removed the baseline difference before microsaccade onset due to the stimulus luminance.

## Direction modulation index

The directional modulation index (DMI) was calculated based on the subtracted directional tuning curves of recording sites. For each site, the directional tuning was calculated by averaging responses during

126–450 ms after microsaccades in different directions relative to the RFs. Then we subtracted baselines (the minimum value) from the tuning curves. Then we calculated the DMI for each site as:

$$\text{DMI} = \frac{(R_{-45°} + R_{0°} + R_{45°}) - (R_{-135°} + R_{180°} + R_{135°})}{(R_{-45°} + R_{0°} + R_{45°}) + (R_{-135°} + R_{180°} + R_{135°})} \quad (1)$$

where $R$ is a response value from the tuning curve removed the baseline.

### Laminar alignment

To align different probe placements in depth, we used the laminar pattern of MUA responses combined with the current source density (CSD) analysis of LFP signals[31–33,36]. The CSD profile can be estimated according to the finite difference approximation, taking the inverse of the second spatial derivative of the stimulus-evoked voltage potential φ, defined by:

$$\text{CSD}(x) = \frac{\varphi(x+h) + \varphi(x-h) - 2\varphi(x)}{h^2} \quad (2)$$

where $x$ is the depth at which the CSD is calculated and h the electrode spacing (100 μm).

### Latency

To calculate the latency of microsaccade direction modulation on population firing rates, we compared firing rates (smoothed) for two direction conditions using a two-tailed $t$ test. The first 20-ms consecutive significant ($p < 0.05$, Bonferroni corrected) time points defined the latency of microsaccade direction modulation. When comparing population latencies across different layers, we used bootstrap method to sample N-1 penetrations for each time which simultaneously recorded activities from all three layers, and then calculated the averaged responses and latencies for each layer. This procedure was repeated 1000 times, and the results were entered into nonparametric ANOVA (Kruskal–Wallis) due to the violation of the normal distribution assumption.

### Behavior analysis

To obtain the time course of the detection accuracy, we first calculated the time interval between the nearest microsaccade generated just before the target in each trial, and ensured that there were no microsaccades generated within 0.2 s after target (avoiding the possible influence of microsaccade preparation on behavior). Then, we computed the mean value of the accuracy in time bins of 40 ms separately across trials (from all sessions) in which the microsaccade before the target was directed toward the target side. The time bin was equivalent to a sampling rate of 25 Hz for temporal dynamics of behavioral performance. We used the bootstrap method to test the statistical significance of the difference between the toward and away conditions. For each time, we sampled N trials in each bin of the microsaccade-target interval and calculated the mean accuracy separately for the toward and away conditions. Repeating this procedure 1000 times yielded an empirical distribution of the accuracy values, allowing us to estimate 95% confidence intervals for the accuracy in each timestep, and to compare the difference in the accuracy of the toward and the away conditions using the bootstrap percentile method.

### The dynamic recurrent model

We built a three-stage dynamic network model in a mean-field approach to simulate neurophysiological and behavioral data. The first two stages were composed of two excitatory neural populations with RFs located in the left or right hemisphere. The first stage ($P$) simulated V1 population activity and was described by the following dynamical equations:

$$\tau_P \frac{dP}{dt} = -P + I_{\text{visual}} + w_S S + \text{noise} \quad (3)$$

The excitatory component $P$ received visual pulse input $I_{\text{visual}}$ in its RF and biphasic microsaccade kernel ($S$) and Gaussian noise (N (0, 0.03)). The visual input ($I_{\text{visual}}$) is a brief current that is held constant during target presentation (20 ms) and the biphasic kernel was a normalized response in V1 from our neurophysiological data. $\tau_P$ is the rate at which the excitatory component ($P$) approaches its steady state (0.02 s). The second stage ($V$) simulated the V2 activity and received output from the corresponding $P$ with RF in the same location and direction modulation ($M$). This was described by the following dynamical equation:

$$\tau_V \frac{dV}{dt} = -V + F[P + M] \quad (4)$$

where $\tau_V$ is the rate at which the excitatory component ($V$) approaches its steady state (0.035 s). $F$ was an activation function defined by:

$$F(x;a,\theta) = \frac{1}{1 + e^{-a(x-\theta)}} + \frac{1}{1 + e^{a\theta}} \quad (5)$$

where the threshold $\theta$ is 2.8, and the gain $a$ is 1. The direction modulation was described by the following dynamical equation:

$$\tau_M \frac{dM}{dt} = -M + I_{\text{direction}} \quad (6)$$

where $M$ received a directional suppressive input ($I_{\text{direction}}$) of brief constant current with two parameters: latency and duration (0.1 s). $\tau_M$ is the rate at which the inhibitory component ($M$) approaches its steady state (0.1 s). $M$ only suppresses the component in the second stage ($V$) with RF in the opposite direction of microsaccades. In the third stage, two excitatory neural populations (selective for left and right target locations) were labeled L and R. The $D_L$ and $D_R$ are given by Eqs. 4 and 5 as follows:

$$\tau_D \frac{dD_L}{dt} = -D_L + F[w_{LL}D_L - w_{LR}D_R + V_L] \quad (7)$$

$$\tau_D \frac{dD_R}{dt} = -D_R + F[w_{RR}D_R - w_{RL}D_L + V_R] \quad (8)$$

The negative sign in front of $w_{LR}$ and $w_{RL}$ indicates that the overall effective connectivity between the two populations is inhibitory. $\tau_D$ is 0.05 s. Parameters of weights tuned in the model: $w_S = 0.15$, $w_{LL} = 5$, $w_{LR} = 5$, $w_{RR} = 5$, $w_{RL} = 5$.

In a simulated trial, the decision period covered the potential target onset time (0.5–2 s), and the population at the stage 3 ($D_L$ or $D_R$) that reached the threshold for the longest time during the decision period would win the left/right decision. We also built a control model by removing the direction modulation ($M$) from the full model. The simulation of all trials was repeated 15 times separately using the full model and the control model. During each simulation, the real onset times of microsaccades and targets and the microsaccade directions in all trials of the behavior task were used as input, which guaranteed that the model could work in an experimental setting.

### Reporting summary

Further information on research design is available in the Nature Research Reporting Summary linked to this article.

## Data availability

The dataset underlying the results are available as downloadable files at https://github.com/yujie1447/NC_Microsaccade. Source data are provided with this paper.

## Code availability

The source code written in Matlab to reproduce results of this study is freely available on GitHub (https://github.com/yujie1447/NC_Microsaccade).

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

## Acknowledgements

This study was supported by the National Natural Science Foundation of China (grant no. 32171033 [D.X.], grant no.32100831 [T.W.]), the Fundamental Research Funds for the Central Universities, The 111 Project Grant (BP0719032), the fellowship of China Post-doctoral Science Foundation (grant no.2021M690435) [T.W.], the BNU Interdisciplinary Research Foundation for the First-Year Doctoral Candidates (BNUXKJC1909) [Y.W.], the Open Research Fund of the State Key Laboratory of Cognitive Neuroscience and Learning. All the above funders had no role in study design, data collection and analysis, decision to publish, or preparation of the manuscript.

## Author contributions

Y.W. and D.X. designed the research. Y.W., T.W., and T.Z. trained the monkeys. T.W., Y.L., Y.W., and Y.Y. performed the surgeries. Y.W. collected the data. Y.W. analyzed the data. Y.W. wrote the original draft. Y.W., T.W., Y.L., Y.Y., W.D., C.H., Y.Z., and D.X. reviewed and edited the paper. D.X. supervised the research.

## Competing interests

The authors declare no competing interests.
