## [Peer Review File · Nature Communications]

V1-bypassing Suppression Leads to Direction-Specific Microsaccade Modulation in Visual Coding and PerceptionREVIEWER COMMENTS

Reviewer #1 (Remarks to the Author):

This interesting paper demonstrates a novel neuronal mechanism underlying the effects of microsaccades on visual perception. It describes a series of carefully-designed and well-executed experiments that will have a major impact in our understanding of the relation between eye movements and vision. The paper first demonstrates that two main areas at early stages of the visual cortical processing (V1 and V2) are differently modulated by small eye movements called microsaccades, which span less than 1 degree in amplitude. They convincingly show that microsaccades modulate the activity of area V2 but not area V1 and that the modulation in area V2 is tuned to microsaccade direction and is strongest when the microsaccades move the eye away from the V2 neuronal receptive field. The authors provide two excellent controls to demonstrate that these V2 activity modulations are not driven by visual stimuli. They then show evidence that the microsaccade response modulation originates at the input layer of V2, is caused by response suppression, and occurs in monkeys that also have their visual perception affected by microsaccade direction. The paper also provides a computational model that reproduces the effect of microsaccades on V2 activity and visual behavior. I have no major criticisms or concerns with the results and main conclusions of the paper. I just have a few suggestions that I hope will help the authors to improve what is already a strong paper.

1. The controls that the authors show in Figure 2 are very convincing. However, the evidence that microsaccade amplitude is not correlated with firing rate could be stronger. For example, it would be helpful to show the correlation as a scatter plot and report the correlation value. This may be important to help understand the origin of the signals driving the microsaccade modulation of V2 activity. The two controls from Figure 2 are very effective at demonstrating that the V2 activity modulations are not driven by visual stimuli. Therefore, the authors are leading their readers to believe that the modulations are driven by attention and/or motor signals. In the discussion, the authors argue that attention is not the main driver of these modulations. Therefore, motor signals appear to be the most likely source of these modulations. However, if the V2 response modulations are driven by motor signals, I would expect to see some correlation between saccade amplitude and firing rate, even if it is weak (perhaps, I would consider including in this analysis not just microsaccades but also saccades with amplitude > 1 deg). This is not a major criticism. The authors may have other reasons to show the control in the way they are showing it and they can decide what is the best approach to make their paper stronger.

2. The authors may want to leave open the possibility that the microsaccade modulation of V2 activity may be related to attention. Attention not only enhances but also suppresses cortical responses. Moreover, even in the absence of a stimulus, the monkey may have a bias towards attending a spatial position where the stimulus has been previously shown multiple times. There is not much to be gained by claiming that the V2 response modulation is not related with attention. The attention effects related to microsaccades may be weak and easy to miss. If it is not an attention effect, the reader is forced to conclude that the effect is driven by some type of motor signal. If it is a motor signal, I would expect to see a correlation between the response modulation and saccade amplitude.

Other minor comments

1. Line 132. 'We found that the averaged V2 responses after microsaccades moved in different directions are well-tuned'. The term 'well-tuned' is unclear here because the term 'direction tuning' was not introduced yet in the text. It may better to write that '... the averaged visual cortical responses were tuned to microsaccade direction in area V2 (...) but not area V1.'

2. Typo in line 201 (and other parts of this text section): 'faciliatory'. Consider changing to '...might be due to facilitation'.

3. Figure 4B. Please, provide a color scale bar or define yellow and blue in the figure legend. The reader needs this information to understand the color code for response facilitation and suppression.

4. Figure 4D. The dots with different shades of gray are difficult to see. Consider using different colors instead (e.g. green, brown and black). What are the vertical lines superimposed on the asterisks? They are not explained in the figure legend.
5. Figure 4C. The differences in latency are not easy to see in this figure panel (they are very clear in Figure 4D panel). Consider showing dotted lines and latency values next to each panel. Consider also adding labels to directly show in the figure what measurements are done in supragranular, granular and infragranular layers.
6. Figure 5B. It would be helpful to report in the figure panel or figure legend the luminance of the target and background in cd/m². The units of target luminance in the x axis are not reported (is this luminance or luminance contrast?).
7. Line 336. Consider writing '... the target that appeared in...'
8. Figure 6A. Define M and S input in figure panel or figure legend.
9. Line 372. Typo. 'V' should be 'V2'.
10. Line 556. Did you use U probes or V probes? Methods reports V probes but the main text reports U probes.

Reviewer #2 (Remarks to the Author):

Microsaccades (MS) are a form of involuntary saccades which are smaller than 1 degree. MS have been shown to have effects on refreshing visual information. However, how MS modulates visual cortex is unclear and limited studies have shown direction specific modulation effect of MS in early visual cortex. This study investigated the MS influence on V1 and V2 neurons in macaque. Surprisingly, they found direction-specific suppression of MS in V2, but not in V1, and appeared first in the middle layer. They further tested the behavioral consequence by XX task and built a hierarchical recurrent model to link neural activity with behavior.

Major comments:

Why would neurons in V2 show significant modulation from MS, while V1 neurons showed no effect, regardless of the strong recurrence between the two areas? It would be helpful to rule out the following possibilities.

1. Could it be due to the RF location (central vs peripheral) of the recorded V1 and V2 neurons? I would like to see the distribution of center of RF for V1 and V2 units relative to fixation.
2. Alternatively, is it because the stimulus size is 4-6 degree, which is much larger than the V1 receptive field size and induced a strong suppression in V1 to begin with? To rule out this factor, it would be helpful to use smaller stimulus size for a control.
3. Based on the experimental design, both V1 and V2 units are simultaneously recorded. However, there are inconsistency of sessions been analyzed in each monkey between V1 and V2. Would this cause a difference in the observed V1 and V2 difference?
4. Is there any animal specific MS modulation effect? I would like to see a supplemental figure plotting the effect of MS on V1 and V2 units in the three macaque monkeys separately. I noticed there are significantly more sessions in DQ with V2 recording than V1 recording, and also more than number of sessions in other monkeys. I wonder whether this will introduce any bias.
5. I also have some doubt about the V1 data quality. Based on the relative power plot of V1 recording sites shown in supplement figure 2, this is no obvious gamma peak given the size of the stimulus (like the peak around 40Hz in V2). I wonder maybe the reason this study didn't observe effect in V1 is due to poor data quality in V1.

It will make the conclusion more solid, if these concerns can be resolved.

The hierarchical recurrent model is a nice integration of both experimental and behavioral results. It would be helpful if the motivation for model design and assumptions can be clearly stated. What component is necessary for this model to realized its prediction, especially the dynamics? How about alternative/control models?

Some additional discussion points worth adding to the current study:

1. Computational importance to have direction-specific microsaccades modulation.
2. Reconciling with the inconsistent literature that didn't report direction-specific MS modulation.

Minor comments:

1. line 65: "we report that direction-specific microsaccade modulation (DSMM) emerges in V2": it is better to use 'observed' instead of 'emerges'
2. The term 'V2 input layer' is confusing, since inputs to V2 can be from either bottom up or top-down. Please be more specific. Replace with middle layer.
3. "suppressive mechanism": the suppressive effect observed in the study belong to observation rather than mechanism

Dear reviewers,

Below are our point-by-point responses to your comments. For easy identification, our responses are written in the style of bold and the corresponding reviewer comments were marked in the style of italics and underline in the file. The changes in the manuscript were highlighted in red color. Your comments were very encouraging and helpful, and we sincerely thank both of you for the constructive feedback. We have made substantial changes for figures, text, figure legends and supplementary information. Addressing the comments has improved and clarified the communication of the findings we report in our manuscript. We hope that our responses and revision are satisfactory to both of you.

Sincerely,
Dajun Xing

Reviewer #1 (Remarks to the Author)

This interesting paper demonstrates a novel neuronal mechanism underlying the effects of microsaccades on visual perception. It describes a series of carefully-designed and well-executed experiments that will have a major impact in our understanding of the relation between eye movements and vision. The paper first demonstrates that two main areas at early stages of the visual cortical processing (V1 and V2) are differently modulated by small eye movements called microsaccades, which span less than 1 degree in amplitude. They convincingly show that microsaccades modulate the activity of area V2 but not area V1 and that the modulation in area V2 is tuned to microsaccade direction and is strongest when the microsaccades move the eye away from the V2 neuronal receptive field. The authors provide two excellent controls to demonstrate that these V2 activity modulations are not driven by visual stimuli. They then show evidence that the microsaccade response modulation originates at the input layer of V2, is caused by response suppression, and occurs in monkeys that also have their visual perception affected by microsaccade direction. The paper also provides a computational model that reproduces the effect of microsaccades on V2 activity and visual behavior. I have no major criticisms or concerns with the results and main conclusions of the paper. I just have a few suggestions that I hope will help the authors to improve what is already a strong paper.

Reply: We appreciate your positive comments and recognition for our study. Thank you very much!

1. The controls that the authors show in Figure 2 are very convincing. *However, the evidence that microsaccade amplitude is not correlated with firing rate could be stronger.* For example, it would be helpful to show the correlation as a scatter plot and report the correlation value. This may be important to help understand the origin of the signals driving the microsaccade modulation of V2 activity. The two controls from Figure 2 are very effective at demonstrating that the V2 activity modulations are not driven by visual stimuli. Therefore, the authors are leading their readers to believe that the modulations are driven by attention and/or motor signals. In the discussion, the authors argue that attention is not the main driver of these modulations. Therefore, motor signals

appear to be the most likely source of these modulations. *However, if the V2 response modulations are driven by motor signals, I would expect to see some correlation between saccade amplitude and firing rate, even if it is weak* (perhaps, I would consider including in this analysis not just microsaccades but also saccades with amplitude > 1 deg). This is not a major criticism. The authors may have other reasons to show the control in the way they are showing it and they can decide what is the best approach to make their paper stronger.

Reply: Thank you for the nice and important suggestion, which helps us further understand neural origin for the directional modulation of microsaccades (MS). Following your suggestion, we did further analysis on the relationship between MS amplitudes and firing rates modulated by MS. In the old version of our manuscript, we only looked at the correlation between MS amplitudes and MS modulation without taking care of directions of these microsaccades. In the new correlation analysis, we separated microsaccades toward and away from RFs to investigate the possible interaction between MS direction and amplitude on modulating firing rates, and we provided result of V1 as a control.

We first showed the scatter plots of four example sites (Figure R1 a-b) where each dot represents a single microsaccade and its modulation on neural response. In these examples, only MSs away from RFs brought significant correlations between MS amplitude and firing rates in V2. However, due to large fluctuations of neural response during each MS, the linear trend was hard to detect and only a few sites exhibited significant correlation. We are unable to include large saccades (larger than 1 degree) in the analysis, because we have restricted the circular fixation window with a radius of 1 degree, and our data collection system discarded a trial if there was any microsaccade larger than 1 degree.

In order to obtain reliable modulation signals, we binned neural responses of each site (in V1 and V2) by amplitudes of all available microsaccades in ‘toward’ and ‘away’ directions respectively, and calculated correlation between mean amplitudes of binned microsaccades and their mean modulations for all sites (Figure R1 c-d). Interestingly, during the late period (126 ~ 450 ms, the period where we found direction modulation in V2), we found a significant correlation for ‘away’ direction but not for ‘toward’ direction (Figure R1d). The difference between the two correlation coefficients was significant ($z = 2.36, p = 0.02$). Such a weak but significant relationship was only observed in V2, but not in V1 (Figure R1c). The results held consistent when we did correlation analysis on the group means of modulation and ten levels of amplitudes (Figure R1e-f). The correlation in ‘away’ direction at late period supports the view that directional suppressive modulation after microsaccades in V2 might be related to motor signals.

We have added the correlation results in main text (in lines 203-214 on pages 10-11) and added the scatter plots (Figure R1e-f) as part of the new figure 3 (Fig. 3d) in the revision, as the evidence supporting the motor source for the direction-modulation of MS in V2. Thank you for the suggestion!

Figure R1 Relationship between microsaccade modulation and amplitude

a Correlation between microsaccade amplitude and averaged neural response during late period (126 ~ 450 ms) from two example sites in V1. **b** Correlation between microsaccade amplitude and averaged neural response during same period from two example sites in V2. **c** Normalized responses of each site (in V1) binned by amplitudes of microsaccades in 'toward' and 'away' directions respectively. Blue and orange dots indicate the mean responses in each amplitude bin. **d** Normalized responses of each site (in V2) binned by amplitudes of microsaccades in 'toward' and 'away' directions respectively. Solid blue line indicates the fitted line. **e** Correlation between averaged V1 response across units and microsaccade amplitude. Solid lines indicate fitted results. **f** Correlation between averaged V2 response across units and microsaccade amplitude. Confidence interval is shown as blue shadow.

2. The authors may want to leave open the possibility that the microsaccade modulation of V2 activity may be related to attention. *Attention not only enhances but also suppresses cortical responses. Moreover, even in the absence of a stimulus, the monkey may have a bias towards attending a spatial position where the stimulus has been previously shown multiple times.* There is not much to be gained by claiming that the V2 response modulation is not related with attention.

The attention effects related to microsaccades may be weak and easy to miss. If it is not an attention effect, the reader is forced to conclude that the effect is driven by some type of motor signal. If it is a motor signal, I would expect to see a correlation between the response modulation and saccade amplitude.

Reply: Thank you for the suggestion. We appreciate the solid logic behind your suggestion. Our response to your first comment suggests that the directional modulation of MS is very likely to come from motor source. However, we do agree with you about that attention not only enhances but also suppresses neural responses to irrelevant information. And we cannot fully exclude the attention effect that is time-locked to microsaccade since we did not manipulate attention itself in the fixation task. Following your suggestion, we have revised the discussion part (in line 437-453 on page 20) to leave open the possibility that the microsaccade modulation of V2 activity may be also related to attention; and we also raised the possibility that we might miss some weak attention effects related to microsaccades.

Other minor comments

1. Line 132. 'We found that the averaged V2 responses after microsaccades moved in different directions are well-tuned'. The term 'well-tuned' is unclear here because the term 'direction tuning' was not introduced yet in the text. It may better to write that '... the averaged cortical responses were tuned to microsaccade direction in area V2 (...) but not in area V1.'

Reply: We have changed the phrase 'well-tuned' to the descriptive sentence '... the averaged visual cortical responses were tuned to microsaccade direction in area V2 (...) but not area V1'. Please see the changes labeled in red color in lines 97-98 on page 6. Thank you for the suggestion!

2. Typo in line 201 (and other parts of this text section): 'faciliatory'. Consider changing to '...might be due to facilitation'.

Reply: We have changed 'faciliatory modulation' to 'facilitation' in main text and figure legends (please see changes in line 174 on page 9, line 460 on page 21 and in line 885 on page 35).

3. Figure 4B. Please, provide a color scale bar or define yellow and blue in the figure legend. The reader needs this information to understand the color code for response facilitation and suppression.

Reply: We apologize for our negligence. A color scale bar has been added in Figure 4B.

4. Figure 4D. The dots with different shades of gray are difficult to see. Consider using different colors instead (e.g. green, brown and black). What are the vertical lines superimposed on the asterisks? They are not explained in the figure legend.

Reply: Thank you for the suggestion. We have changed the color of dots to red, blue and yellow to represent sites from three monkeys. The vertical lines with asterisks denote significant result from post-hoc comparisons of Kruskal-Wallis test (Bonferroni-corrected). We have changed the location of asterisks in Figure 4d, and added the corresponding description in the figure legend (in lines 915 on page 36).

5. Figure 4C. The differences in latency are not easy to see in this figure panel (they are very clear in Figure 4D panel). Consider showing dotted lines and latency values next to each panel. Consider also adding labels to directly show in the figure what measurements are done in supragranular, granular and infragranular layers.

Reply: We defined latency by the first bin of time where there is a significant difference of responses between toward and away conditions in paired t test (Bonferroni-corrected). The time point with significant difference in the two directions has been shown as gray bars in Figure 4c, and we have added vertical lines in each panel to show the latency in each layer. The latency values were added in the figure legend (please see line 908-909 on page 36).

6. Figure 5B. It would be helpful to report in the figure panel or figure legend the luminance of the target and background in cd/m². The units of target luminance in the x axis are not reported (is this luminance or luminance contrast?).

Reply: Thank you for the suggestion. The x axis now is labelled as contrast (defined as $(\text{luminance}_{\text{(target)}} - \text{luminance}_{\text{(background)}}) / \text{luminance}_{\text{(background}})$). We added the physical luminance value in cd/m² in the figure legend (please see lines 926-927 on page 36).

7. Line 336. Consider writing ‘... the target that appeared in...’.

Reply: Thank you for the suggestion. We have changed ‘... the target appeared in ...’ to ‘... the target that appeared in ...’ in line 289 on page 15.

8. Figure 6A. Define M and S input in figure panel or figure legend.

Reply: We apologize for the unclear statement of the model elements. To help readers’ understanding, we replaced S with ‘V1 biphasic response’ which took input of onset times of microsaccades. We have added the description of the model components (M and other components) in the figure legend (please see lines 937-943 on page 37).

9. Line 372. Typo. ‘V’ should be ‘V2’.

Reply: We would like to clarify that ‘V’ here represent the component V in the model stage 2, since the panel shows simulation results of mean-field activities after averaging V_L and V_R according to microsaccade direction. We have changed ‘V’ to ‘ V in stage 2’ in line 950 on page 37 and changed the title of the panel (Figure 4e) to ‘stage 2’ to avoid misinterpretation. We also changed ‘ V ’ to ‘the stage 2 (V)’ in the other places in the text for the same reason. Thank

you!

10. Line 556. Did you use U probes or V probes? Methods reports V probes but the main text reports U probes.

Reply: We apologize for the mistake. We used V probes and we have changed 'U' to 'V' in line 61 on page 4.

Reviewer #2 (Remarks to the Author):

Microsaccades (MS) are a form of involuntary saccades which are smaller than 1 degree. MS have been shown to have effects on refreshing visual information. However, how MS modulates visual cortex is unclear and limited studies have shown direction specific modulation effect of MS in early visual cortex. This study investigated the MS influence on V1 and V2 neurons in macaque. Surprisingly, they found direction-specific suppression of MS in V2, but not in V1, and appeared first in the middle layer. They further tested the behavioral consequence by XX task and built a hierarchical recurrent model to link neural activity with behavior.

Major comments:

Why would neurons in V2 show significant modulation from MS, while V1 neurons showed no effect, regardless of the strong recurrence between the two areas? It would be helpful to rule out the following possibilities.

Reply: Thank you for the constructive suggestions, which help us rule out several possible V1 sources for the directional microsaccade (MS) modulation in V2. We have included more experimental data and performed further analyses to address each possibility you suggested. Results in responding to both your questions below and the question from the reviewer #1 (Figure R1) suggest that the directional modulation of microsaccades in V2 is more likely to come from motor system, and it is unlikely to directly come from V1.

You have also indicated a very interesting question, that is since there is a feedback loop (recurrence) from V2 to V1, why directional modulation by MS in V2 doesn't feedback to V1? We think this is mainly due to the following reasons. Anatomical studies (Rockland & Pandya, 1979; Vanni, Hokkanen, Werner, & Angelucci, 2020) have shown that the feedback loop from V2 to V1 is laminar specific, mainly between the deep layer of V2 and the superficial layer of V1 (Figure R2a). Based on the laminar pattern of directional modulation in V2 shown in Figure 4d (we also copy this figure as Figure R2b), the directional modulation of MS in the deep layer of V2 was significantly weaker than that in V2 middle layer. Therefore, V2 feedback (from V2 deep layer) effect on V1 might not have strong direction modulation of MS. This might explain why our results don't show significant direction modulation of MS in V1.

Figure R2 Recurrence between V1 and V2 and laminar profile of direction modulation strength

a Schematics of laminar circuits between V1 and V2. **b** Laminar distribution of direction modulation index in V1 and V2. The vertical black and cyan lines indicate the mean of each depth bin for each cortical region. Shaded regions denote the SE across units.

It is worth pointing out that, we did not see significant direction modulation of MS in V1, but we did see strong and significant non-direction modulation of MS in V1 (biphasic response in figure 1d), which is consistent with existing literatures (Hass & Horwitz, 2011; Kagan, Gur, & Snodderly, 2008; Meirovithz, Ayzenshtat, Werner-Reiss, Shamir, & Slovin, 2012). And the laminar pattern of biphasic modulation in V1 (figure 4b) was highly consistent with a previous study (McFarland, Bondy, Saunders, Cumming, & Butts, 2015): the suppression first appeared in the granular and infra-granular layers of V1, specifically layers 4 and 6. Compared to the strong non-directional biphasic modulation in V1, which indicates good V1 response, the directional effect in V1 was very weak. We speculate that strong non-directional modulation of MS might mask weak directional modulation of MS in V1 (due to feedback from V2). In the superficial layer of V1, we do identify a very weak trend of direction modulation of MS (L2/3: $t_{101} = 1.95$, $p = 0.054$, two-sided t test), which might indicate some recurrence from V2 to V1, although the direction modulation of MS is not significant in any cortical layers.

Overall, we think V1 might not be a major target of directional modulation by MS. Our results suggest that directional modulation is mainly from a motor source projecting to V2 middle layer and have a strong impact on downstream visual processing while show very weak effect in V1 via V2-V1 feedback modulation. Thank you for this comment, and we have added more discussion about this point in main text (please see lines 388-402 on page 18-19).

To check whether V1 have significant directional modulation of MS, we have carefully inspected all possibilities you suggested, by performing further analyses and including more experimental data elaborated below.

1. Could it be due to the RF location (central vs peripheral) of the recorded V1 and V2 neurons? I would like to see the distribution of center of RF for V1 and V2 units relative to fixation.

Reply: Thank you for the suggestion. We have added the distribution of RF centers in Figure 1b (main text: lines 64-65 on page 4) and explained the result here in more details. We estimated/mapped the receptive field of each unit in our study with small black/white squares rapidly (20 ms) and randomly presented at different location on the screen (Sparse Noise Experiment, Figure R3a, see method for details, lines 579-586 on page 26). We then used a two-dimensional gaussian function here to estimate the center and spatial profile of receptive fields (Figure R3b and c).

Figure R3 The distribution of RF for V1 and V2

- The trial procedure in the sparse noise experiment. Red dashed circle indicates the RF location of a recorded unit. The red dot represents the fixation point.
- RF fitting for one example unit in V1 and V2. Red circles indicate the RF with diameter of 4σ of fitted 2D gaussian.
- The spatial distribution of center of RFs for V1 and V2 units.
- The distribution of RF eccentricity for V1 and V2 units.

The spatial distribution for centers of RFs for V1 and V2 units, relative to fixation point (0,0) was shown in Figure R3c. The RFs of most units from V1 and V2 overlapped with each other and located within the eccentricity of 5° . The median of RF eccentricity of V1 units was 3.05° and the median of RF eccentricity of V2 units was 3.67° . And the difference of the two eccentricity medians (0.62 deg) was smaller than the size (diameter) of RFs of V2 neuron ($0.8\sim 1.2 \text{ deg}$). In our responses to your 3rd comment, we will further demonstrate that the eccentricities of V1 and V2 sites in our study have a negligible effect on the results (Figure R7).

2. Alternatively, is it because the stimulus size is 4-6 degree, which is much larger than the V1 receptive field size and induced a strong suppression in V1 to begin with? To rule out this factor, it would be helpful to use smaller stimulus size for a control.

Reply: Thank you for the nice suggestion of a control experiment with small size. We set stimulus size at $4^\circ\sim 6^\circ$ in the experiments, to guaranteed that RFs of V1 and V2 were fully covered by the surface of visual stimuli, and RFs would not be directly driven by the edge of a stimulus even with the consideration of RF shifts caused by microsaccades ($<1^\circ$). Such an experimental design made the response dynamic after microsaccade purely reflect a motor signal, without changing the visual stimulus within RFs of V1 and V2. To better rule out the possibility that surround suppression might reduce the directional modulation of MS in V1, as you recommended, we used some smaller sizes in a control experiment on monkey DQ and

DS. We recorded V1 neurons on DQ by a Utah array (Figure R4a) with a uniform disk shown in the center of RFs for 2 seconds. We found that surround suppression can change the amplitudes of non-directional modulations by MS; but there was no significant directional modulation of MS in V1, even when smaller sizes of stimulus was used ($<0.1^\circ$, $0.2\sim 0.5^\circ$, 2.3° ; $p > 0.05$, *paired t test* with corrections for multiple comparisons). With data recorded in the V1 of monkey DS by a linear array (V-Probe), we still did not find significant directional modulation with smaller stimuli ($<0.1^\circ$, $0.3\sim 0.6^\circ$) (Figure R4f-g). Based on the results of the control experiment on two monkeys, we hope you are convinced that the possibility that surround suppression in V1 impairs the potential directional modulation after microsaccade can be ruled out. We have added the results of the control experiment in supplementary information (Supplementary Fig. 9) and also mentioned in the discussion (please see lines 396-398 on page 19).

Figure R4 The control experiment with small size in V1.

- V1 recording with a Utah array in monkey DQ.
- Normalized MUA in V1 with a small circle ($<1^\circ$) represent on the center of RFs (orange and blue lines in the left vertical axis) and averaged velocity of eye movement (the black line in the right vertical axis) around microsaccade onset.
- Same with b but the stimulus size was $0.2\sim 0.5^\circ$.
- Same with b and c but the stimulus size was 2.3° .
- Laminar recording in V1 in monkey DS.
- Same with b in monkey DS.
- Same with f but the stimulus size was $0.3\sim 0.6^\circ$

We provided another way to test whether the difference of V1 and V2 on direction-specific microsaccade modulation is due to response excitation/suppression (caused by factors such as surround suppression). We checked whether direction-specific microsaccade modulation is different for recording sites at different response levels, by analyzing the relationship between

direction modulation index and baseline firing rates before microsaccades. We did not find any significant correlation in either V1 ($r = 0.07$, $p = 0.199$) or V2 ($r = 0.11$, $p = 0.051$). We further divided all channels into 5 groups based on quintiles of their baseline firing rates before microsaccades (Figure R5), and we did not find significant directional modulation of microsaccades in any level of baseline firing rates in V1. In V2, all groups exhibited significant directional modulation. This result indicates that the level of excitation/suppression during microsaccade generation in visual cortex might not influence the directional modulation strength. We have added the results of control for baseline firing rates in Figure 2b and in the main text (please see lines 149-156 on page 7-8). Thank you for the advice!

Figure R5 Correlation between baseline firing rates and direction modulation index
a Distribution of normalized baseline firing rates before microsaccades of V1 and V2 channels. **b** Direction modulation index of V1 and V2 populations in each baseline firing rates groups defined by five quintiles.

3. Based on the experimental design, both V1 and V2 units are simultaneously recorded. However, there are inconsistency of sessions been analyzed in each monkey between V1 and V2. Would this cause a difference in the observed V1 and V2 difference?

Reply: We apologize for not describing the experimental design clearly. The 24-channel probe (~2.66 mm) was not long enough to simultaneously record full layers in both V1 (~2 mm) and V2 (~1.3mm). We have to record one region at a time to guarantee the whole laminar was covered. The recording sessions for V1 and V2 were alternated in different days due to the limited trial number that monkeys could bear in one day. That's why the sessions numbers in V1 and V2 for each monkey is not the same. We have added a brief description of the recording method in the main text (in lines 62-64 on page 4).

Among the 32 sessions of V2 recordings, there were 18 sessions that all layers of V2 and part of the above V1 (middle and deep layers) were simultaneously recorded by a single probe.

One example penetration and the relative RF locations of V1 and V2 units were shown in Figure R6a. In these sessions, the stimulus was large enough to cover RFs of all recorded units (in both V1 and V2). These V1 units ($n = 104$) were not used in the old version of manuscript but we included them into the analysis for simultaneous recording in V1 and V2 as a control for the session number. We found the modulation strength in V1 after microsaccade with two directions was comparable, while in V2 firing rates after microsaccades towards RFs were significantly higher than firing rates after microsaccades away from RFs (Figure R6b). And the population distribution of directional modulation index (DMI) for V2 units was higher than DMI distribution for V1 units (Figure R6c). The results of simultaneous-recording sessions with a single probe were consistent with our main findings from separate recordings in V1 and V2 shown in Figure 1. We have replicated the findings in Figure 1, 2 and 3 by using the simultaneous-recording data, which were shown in the supplementary information (Supplementary Fig. 5) and were mentioned in the main text (in lines 117-124 on pages 6-7, lines 147-149 on page 7, lines 194-202 on page 10). Thank you for the question!

Figure R6 Simultaneous recordings in V1 and V2

- Simultaneous recording in V1 and V2 and the RF mapping for each channel from an example session.
- Normalized MUA in V1 and V2 with a square represent on the center of RFs of V1 and V2 units (orange and blue lines in the left vertical axis) and averaged velocity of eye movement (the black line in the right vertical axis) around microsaccade onset. Gray bar on the top indicates the significant time period in the *paired t*-test with Bonferroni corrections.
- Population distribution of direction modulation index of each channel in V1 and V2. '****' indicates $p < 10^{-10}$ and 'ns' indicates $p > 0.05$ in two-sided *t*-test.

Your question also enlightened us to provide another control analysis for comparing directional modulation by MS in V1 and V2 with overlapping RFs but at different sessions of probe recordings from the same monkeys. Usually overlapping RFs in V1 and V2 indicate the recording column in V2 receive direct feedforward projections mainly from the corresponding

column in V1, which can help to dissect the input from V1 and new processing mechanism in V2 without the potential confounding of column diversity. A potential drawback for simultaneous recording by a single probe was that the RFs of V1 and V2 don't overlap due to the folding structure of cortex. Considering this point, we selected the sessions with overlapping RFs in V1 and V2 within each monkey (Figure R7a) and did the same analysis as Figure 1 to compensate the drawback of simultaneous recording of one probe. We found similar results that V2 exhibited directional modulation from time course of firing rates, direction tuning and population distribution while the modulation in V1 did not show significant difference between toward and away conditions (Figure R7). We have added the results from units with overlapping RFs in supplementary information (Supplementary Fig. 8) and also in the discussion (lines 394-396 on page 19).

Figure R7 microsaccade modulation in V1 and V2 with overlapping RFs

- RFs of V1 and V2 units were overlapping from 18 V1 sessions and 12 V2 sessions.
- Normalized MUA (orange and blue lines in the left vertical axis) in V1 (top) and V2 (down) and averaged velocity of eye movement (the black line in the right vertical axis) around microsaccade onset. Gray bar on the top indicates the significant time period in the *paired t*-test with Bonferroni corrections.
- Distribution of direction modulation index in V1 and V2.

We hope to further answer your question by using the simultaneous recording data (7 sessions) from another project/experiment on monkey DS (Figure R8a). In the new experiment, we use two probes to simultaneously record V1 and V2 units with overlapping RFs (Figure R8c). We again found similar results as we shown in figure 1, that is V1 did not show directional modulation by MS while V2 did (Figure 8d-e). Considering the two kinds of simultaneous recording with the same session number in V1 and V2, we hope you are satisfied with our answers.

Figure R8 Simultaneous laminar recordings with two probes in V1 and V2 in monkey DS

- Simultaneous laminar recording in V1 and V2 with two linear probes.
- Current source density pattern in V1 and V2 after stimulus onset.
- The overlapping RFs of 11 units from V1 and 11 units from V2 in one example session.
- Normalized MUA in V1 and V2 (orange, MS toward; blue, MS away) and averaged velocity of eye movement (the black line in the right vertical axis) around micro-saccade onset. There were 7 sessions in total. Gray bar on the top indicates the significant time period in the *paired t*-test with corrections for multiple comparisons.
- Population distribution of direction modulation index of each channel in V1 and V2. **** indicates $p < 10^{-10}$ and 'ns' indicates $p > 0.05$ in two-sided *t*-test.

4. Is there any animal specific MS modulation effect? I would like to see a supplemental figure plotting the effect of MS on V1 and V2 units in the three macaque monkeys separately. I noticed there are significantly more sessions in DQ with V2 recording than V1 recording, and also more than number of sessions in other monkeys. I wonder whether this will introduce any bias.

Reply: Thank you for the suggestion. Results from the three monkeys separately are shown in Figure R9 and we have added the figure into supplemental materials (Supplementary Fig. 2) and added the description in the main text (in lines 92-94 on page 6). Results from each individual monkey are consistent with what we reported by combining data from all three monkeys. As Figure R9 shows, firing rates in V1 after microsaccades in the toward and away conditions showed comparable strength of modulation ($p > 0.05$, *paired t*-test with corrections) for all three monkeys, while firing rates after microsaccades toward RFs were higher than those away from RFs ($p < 0.05$, *paired t* test with corrections for multiple comparisons) in V2 for all three monkeys.

To further test whether the session number of V2 recording on monkey DQ (17 sessions) brought any bias when we comparing V1 and V2 on monkey DQ, we randomly select 5 sessions from 17 sessions and showed the MS effect in Figure R9d. We found even when the number of channels (N = 48) in V2 selected sessions was less than the number in V1 (N = 52, 6 sessions),

the directional modulation of microsaccades was still significant ($p < 0.05$, *paired t test* with corrections). Next, to test the result consistency of subset of V2 sessions from DQ and whether the number of V2 recording sessions from DQ brought any bias on direction modulation effect in V2, we use bootstrapping method to compare direction modulation strength of DQ to two other monkeys. We randomly sampled 8 sessions from 17 sessions (comparable to 9 V2 sessions in DS and 7 V2 sessions in DK) and calculated the average direction modulation index of the subset of data. This procedure was repeated for 200 times and the distribution of means of direction modulation index was shown in Figure R9e. All means of subsets from DQ V2 data were above zero, indicating a consistent and robust directional modulation in V2 of DQ. Based on the percentile test, we did not find any significant difference between the direction modulation strength (DMI) of subsets of DQ V2 sessions and modulation strength of V2 population from DS ($p > 0.05$) or any difference between DMI of subsets of DQ and DMI of V2 population from DK ($p > 0.05$). Furthermore, all three monkeys exhibited significant direction modulation of V2 in a population level ($p < 0.001$, one-sample *t test*).

Based on the above analysis, data from monkey DQ DK and DS all showed significant and comparable strength of direction-specific microsaccade modulation in V2, but only non-directional modulation by MS can be found in V1. This suggests that different numbers of sessions from the three monkeys were unlikely to cause a bias for estimating the direction modulation in V1.

Figure R9 Microsaccade modulation in three monkeys

- Normalized MUA (orange and blue lines in the left vertical axis) in V1 (top) and V2 (down) from monkey DQ and averaged velocity of eye movement (the black line in the right vertical axis) around microsaccade onset. Gray bar on the top indicates the significant time period in the *paired t-test*.

- b. Normalized MUA in V1 and V2 from monkey DK and averaged velocity of eye movement around microsaccade onset.
- c. Normalized MUA in V1 and V2 from monkey DS and averaged velocity of eye movement around microsaccade onset.
- d. Normalized MUA and velocity of eye movement from 5 randomly selected session of monkey V2 recordings.
- e. The distribution of means of direction modulation index (DMI) for bootstrapped sessions. Vertical solid lines represent the means of DMI for all sessions from DS (yellow) DK (blue) and DQ (red). Asterisks donates statistical significance ($p < 0.001$).

5. I also have some doubt about the V1 data quality. Based on the relative power plot of V1 recording sites shown in supplement figure 2, this is no obvious gamma peak given the size of the stimulus (like the peak around 40Hz in V2). I wonder maybe the reason this study didn't observe effect in V1 is due to poor data quality in V1.

Reply: The broad gamma peak in V1 (in Supplementary Fig.6) is due to short time window (150–450 ms after microsaccades) available for us to conduct power spectrum analysis on the LFP after a microsaccade, which is much narrower than that used in traditional spectrum analysis of the LFP (usually longer than 1s). When we use longer time window (0.5s-3s after stimulus onset) to perform spectrum analysis on the LFP from the same dataset, we can get a much clearer gamma peak in V1 (Figure R10), which is consistent with previous studies (Xing, Yeh, Gordon, & Shapley, 2014). To further test whether the absent of direction-specific microsaccade effect in V1 is due to poor quality of the V1 data in some channels, we directly inspected the relationship between directional modulation of MS and data quality of spike activity from recording sites, defined as signal-noise ratio (SNR), in V1 (Figure R11).

Figure R10 LFP power spectrum for different surface luminance in V1 and V2

- a. LFP power spectrum in V1 after stimulus onset (0.5 – 3s).
- b. LFP power spectrum in V2 after stimulus onset (0.5 – 3s).

The signal-noise ratio (SNR) of spike activity for each recording site was defined by the site's relative visual response and its baseline activities:

$$SNR = \frac{\text{mean}(MUA_{\text{initial visual response in } 40-90\text{ms}}) - \text{mean}(\text{baseline MUA})}{\text{std}(\text{baseline MUA})}$$

We defined a channel as a good channel, if the site's SNR was higher than 0.65; in this way we got 278 good channels in V1 and 283 in V2 as shown in Figure R11a. Then we did the same analysis as that for Figure 1 and the results were similar to our findings with the whole dataset: in V1, no significant difference on firing rates after microsaccade toward and away from RFs was found in time course ($p > 0.05$, *paired t-test* with multiple corrections) or spatial tuning; but in V2, significant directional modulation of firing rates after microsaccades and bell-shaped microsaccade direction tuning were found (Figure R11b). When we change the criteria for defining good channels (from SNRs >0.65 to SNRs > 2.5), the above conclusions can still be held (Figure R11d-f). The mean of direction modulation index of the neural population in V1 is again not significant different from zero.

Figure R11 Microsaccade modulation on MUA in V1 and V2 with good channels

- Normalized firing rates around microsaccade in V1 and V2 after selecting channels with SNR > 0.65 .
- Microsaccade direction tuning for the averaged responses of V1 and V2 from the period 125–450 after microsaccades, red and blue shadow regions indicate two direction bins used for calculated direction modulation index in **c**.
- Distribution of direction modulation strength in V1 and V2.
- Same as **a** but the units were selected by the criteria of 2.5.
- Same as **b** for units with SNR over 2.5.
- Same as **c** for units with SNR over 2.5.

We also analyzed the LFP of the channels with good quality as we did in Figure R11 to confirm the findings about gamma (Figure R12). In V1, we found no significant difference between gamma power (50-85 Hz) after microsaccades towards and away from RFs and we did not find a significant higher-than-zero distribution of direction modulation index of gamma (Figure R10b, 0.02 ± 0.30 , $M \pm SD$, $t(277) = 1.14$, $p = 0.26$). In V2, the gamma-band power was higher after microsaccades moved in the toward direction than after microsaccades moved in the away direction (0.20 ± 0.26 , $M \pm SD$, $t(282) = 12.56$, $p < 0.001$). When we change the criteria for defining good channels (from SNRs >0.65 to SNRs >2.5), the above conclusions can still be held (Figure R12c-d). Both MUA and LFP results hold consistent after we chose the good channels.

Figure 12 Microsaccade modulation on LFP in V1 and V2 with good channels

- Relative power spectrum (to blank) of LFP in V1 and V2 with units of SNR over 0.65.
- Distribution of direction modulation strength of gamma power in V1 and V2. '****' indicates $p < 10^{-5}$ and 'ns' indicates $p > 0.05$ in two-sided t -test.
- Same as a but for units with SNR over 2.5.
- Same as b but for units with SNR over 2.5.

We further divided our dataset into 5 bins based on quantiles of signal noise ratio (Figure R13), in order to check the relationship between signal quality and direction modulation. We found in V2, groups in all bins at different level of SNRs exhibited significant direction modulation of microsaccades ($ps < 0.001$, after corrections for multiple comparisons) while in V1 we again did not find any significant direction modulation of microsaccade even in the bin with highest SNRs ($ps > 0.05$, after corrections for multiple comparisons). We also tried SNR calculated based on the sustained response in a longer time window where microsaccades were selected (500-2600ms after stimulus onset) and the results were similar and consistent (V2: $ps < 0.001$; V1: $ps > 0.05$). The correlation between direction modulation index and SNR also doesn't show any significance either in V1 or V2 (V1: $r = -0.00$, $p = 0.99$, $N = 358$; V2: $r = 0.09$, $p = 0.10$, $N = 329$). These results indicate that the data quality evaluated by visual response has little influence on the directional modulation of microsaccade. Considering the consistent directional modulation in V2 across three monkeys, we think the signal quality of current data might not introduce bias into results for V1. We have reported results for the control of signal noise ratio in the main text (please see lines 112-116 on page 6) and added them in the supplementary information (Supplementary Fig. 4-6). Thank you for the advice!

Figure R13 Direction modulation index of V1 and V2 was consistent in different levels of signal noise ratio
a Distribution of signal noise ratio for all units in V1 and V2. **b** Direction modulation index of V1 and V2 populations in SNR groups defined by quintiles. *** indicates $p < 0.001$ and **** indicates $p < 10^{-5}$ in two-sided t -test.

It will make the conclusion more solid, if these concerns can be resolved.

Reply: Thank you very much for raising the above concerns which really help us check our data carefully and provide more solid evidences to support our findings. We hope our responses are satisfactory to you.

The hierarchical recurrent model is a nice integration of both experimental and behavioral results. *It would be helpful if the motivation for model design and assumptions can be clearly stated. What component is necessary for this model to realized its prediction, especially the dynamics? How about alternative/control models?*

Reply: Thank you for the wonderful comment and suggestion. We have added more texts to clearly state the motivations for model design and assumptions for model simulation in main text (lines 296-303 on pages 15). Briefly, we'd like to express the idea that given that V2 units exhibited significant directional microsaccade modulation in the middle layer which receives V1 feedforward inputs, we hypothesized that a spatial-specific projection targeting in V2 could lead to biased visual sensitivity of monkeys' behavior in a visual sensitivity task. To test whether such selective input to V2 would be sufficient for interpreting behavioral dynamic after microsaccades, we constructed a hierarchical dynamic neural network which could make decision in a 2AFC task based on dynamic responses in V2. The model was composed of two feedforward stages (corresponding to V1 and V2) and a recurrent decision stage which was inspired by the simplified decision model that was well defined and widely used to explain the behavioral choice in reaction time tasks (Figure R14a).

We first validated our model structure (without any microsaccade modulation components) in the current detection task by replicating contrast-tuning curve measured psychophysically (Fig. 5b). When changing the magnitude of visual input, the accuracy curve of hierarchical dynamic model could be well fitted by a Naka-Rushton function (Fig. 14b) which shows consistent pattern with monkeys' behavior (Fig. 5b). The result indicates good validity of model structure on the 2AFC detection task. Then, we added microsaccade-related biphasic modulation signal to stage 1 and directional suppressive signal to stage 2 (simulating

V2) based on our experimental findings to investigate their contributions on the directional microsaccade effect in behavior dynamic (full model).

Your suggestion of a control model is a good point to contrast contributions of different components. We have added a control model by removing directional suppressive component *M* from the full model. In order to test whether the detection performance of the control model and the full model is also directionally impaired by microsaccade after microsaccades, we chose the input strength generating a correct rate at 75% in the task (Figure R14b) as what we did in a real monkey experiment, and then simulated all behavioral sessions in a trial-by-trial basis for 15 times. We averaged the simulation results and found that the performance of a control model exhibited decreased accuracy during microsaccade generation without any directional effect (Figure R14d). In contrast, when a directional specific suppression was introduced to the stage 2 of the model (the full model), the directional modulation emerged after microsaccades (Figure 14e). The comparison between the control model and full model indicates that the component *M* (directional suppression in V2) may contribute the directional modulation on behavior dynamics after microsaccades. We have added the results of the control models in Figure 6 and in the main text (lines 340-348 on page 17)

Figure R14 Performance of the full model and control model on predicting behavior dynamics

a Illustration for the structure of the dynamic recurrent network. The model is composed of two visual stages (P and V) that receive visual input, a mutual inhibited decision stage, and two modulation components (biphasic kernel and M) that take input of eye movements (including microsaccade time and direction). P was stimulated by a 20-ms impulse current when there is a visual target in the RF. Each microsaccade generation will trigger a biphasic modulation on both components in stage 1 and impulsive suppression in M . M only suppresses V with RF in the opposite direction of

microsaccades. **b** Mean accuracy (across 30 simulations, each simulation was composed of one repetition of all possible target location and target onset time) for each level of visual input strengths. Error bars represent standard deviation of accuracy across simulations. **c** Change in detection accuracy in monkey DN and DS as a function of target onset time relative to a microsaccade. The red vertical line indicates the onset time of microsaccades. The time courses were evaluated with bootstrap tests and hypothesis tests using the percentile method. Asterisks denote statistical significance ($p < 0.05$ after corrections for multiple comparisons). **d** Averaged simulated behavior accuracy of control model after removing component M across 15 simulations (all trials from all empirical sessions from one monkey compose one simulation). Shaded region represents the standard deviation across simulations. **e** Averaged simulated behavior accuracy of the full model across 15 simulations.

Some additional discussion points worth adding to the current study:

1. Computational importance to have direction-specific microsaccades modulation.

Reply: The suggestion is important for us to understand the functional meaning of our finding. In the revision, we have provided some further insights on the computational importance (lines 411-436 on pages 19-20), and we showed them here.

First, the presence of information about microsaccade direction in visual cortex may be important for visual stability which requires knowledge of eye movements. The brain needs to know the direction of the eye movement to avoid misinterpretation of moving environment during microsaccades (Cavanaugh, Berman, Joiner, & Wurtz, 2016). Previous anatomical studies have found that signals from oculomotor system to V2 relies on topographic projections (Adams, Hof, Gattass, Webster, & Ungerleider, 2000; Benevento & Rezak, 1976; Rezak & Benevento, 1979), which might be the neural basis for direction information in visual cortex. The directional suppression in V2 and in the downstream area could be used for computing and inferring whether the information updating in the brain was due to a rightward microsaccade or leftward move in visual space. The hypothesis was consistent with results in our control experiment of the moving surface (Figure 2d), which showed V2 neurons could reveal the difference between the object moving and the microsaccades.

Second, the directional modulation might help the visual cortex reset excitability. The continuous changes of suppression strength in cortical space of V2 after microsaccades result in unbalanced cortical excitability. For a subpopulation whose RFs are in the opposite direction of microsaccades, the feedforward visual stream from V1 needs to be stronger to counterbalance the suppression, while the visual stream to another subpopulation with RFs in the direction of microsaccades more easily reached the threshold. We found that gamma-band activity, which represent the processing of feedforward visual information in visual cortex (van Kerkoerle et al., 2014), was also modulated by microsaccade direction. The unbalanced strength when driving downstream cortical regions leads to a reset of spatial visual sensitivity after each microsaccade.

2. Reconciling with the inconsistent literature that didn't report direction-specific MS modulation.

Reply: Thank you for the suggestion. This is also an important point to discuss. V2 area of the monkey is hard for recordings, because most of V2 area is not on the surface of the brain

and they are underneath V1 area (see figure 1a for demonstration). Therefore, only three studies, to our best knowledge, have investigated microsaccade modulation in V2 (Leopold & Logothetis, 1998; Meirovithz et al., 2012; Tse, Baumgartner, & Greenlee, 2010). The neurophysiology study (Meirovithz et al., 2012) by voltage-sensitive dye imaging found that microsaccades induced monophasic response to small stimuli and induced biphasic response to large stimuli. The study of single cell recording (Leopold & Logothetis, 1998) also found strong monophasic enhancement in V2 following microsaccade when presenting a small grating in the RFs. And in the human functional MRI study, microsaccade induced an increase in BOLD signal peaking at 4.5 seconds then followed by a decrease below baseline with a large polar grating in the screen (Tse et al., 2010). However, none of them has focused on the question of how microsaccade modulates neural activity in V2 in a directional manner, which reflect the new perspective of current study. In addition to the lack of analysis on directional specific modulation by microsaccades in V2, the displacement of a pattern stimulus in RFs due to microsaccades also makes it difficult to determine the direction modulation from the extra-retina source, which is also the case for studies in the middle temporal (MT) region (Bair & O'Keefe, 1998; Herrington et al., 2009).

Our findings in V2 indicate the importance of microsaccade direction relative to RFs for determining its effect on neural activity, which is nonnegligible when trying to explaining the behavioral benefit of microsaccade toward a visual target. One study (Zanos, Mineault, Nasiotis, Guitton, & Pack, 2015) found the amplitude of travelling wave in V4 was larger after large saccade toward RFs, which is consistent with the current perspective in our study. And a more recent study of microsaccade found that only microsaccades toward RFs will follow attentional enhancement on firing rates of V4 neurons (Lowet et al., 2018). These two studies together with our current findings support a new view when taking microsaccade direction into consideration. The recent studies (Lowet et al., 2018; Zanos et al., 2015) on directional modulation of microsaccade in V4 area also indicate that there hasn't been any similar study in V2, the upstream area for V4.

We have added more texts to discuss our findings in V2 regarding to existing literatures (please see lines 371-387 on page 18, lines 403-410 on page 19). Thank you for the suggestion.

Minor comments:

1. line 65: "we report that direction-specific microsaccade modulation (DSMM) emerges in V2": it is better to use 'observed' instead of 'emerges'

Reply: We have changed the word 'emerges' to 'observed' in line 49 on page 3. Thank you for the suggestion.

2. The term 'V2 input layer' is confusing, since inputs to V2 can be from either bottom up or top-down. Please be more specific. Replace with middle layer.

Reply: we have changed the 'V2 input layer' to 'middle layer of V2' in line 18 on page 2 and line 216 on page 11.

3. "suppressive mechanism": the suppressive effect observed in the study belong to observation rather than mechanism

Reply: Thank you for the suggestion. we have changed the word ‘mechanism’ to ‘effect’ in line 50 on page 3 and changed ‘suppressive mechanism’ to ‘suppression’ in line 296 on page 15.

Reference

- Adams, M. M., Hof, P. R., Gattass, R., Webster, M. J., & Ungerleider, L. G. (2000). Visual cortical projections and chemoarchitecture of macaque monkey pulvinar. *J Comp Neurol*, *419*(3), 377-393. doi:10.1002/(sici)1096-9861(20000410)419:3<377::aid-cne9>3.0.co;2-e
- Benevento, L. A., & Rezak, M. (1976). The cortical projections of the inferior pulvinar and adjacent lateral pulvinar in the rhesus monkey (*Macaca mulatta*): an autoradiographic study. *Brain Res*, *108*(1), 1-24. doi:10.1016/0006-8993(76)90160-8
- Cavanaugh, J., Berman, R. A., Joiner, W. M., & Wurtz, R. H. (2016). Saccadic Corollary Discharge Underlies Stable Visual Perception. *J Neurosci*, *36*(1), 31-42. doi:10.1523/JNEUROSCI.2054-15.2016
- Hass, C. A., & Horwitz, G. D. (2011). Effects of microsaccades on contrast detection and V1 responses in macaques. *J Vis*, *11*(3), 1-17. doi:10.1167/11.3.3
- Kagan, I., Gur, M., & Snodderly, D. M. (2008). Saccades and drifts differentially modulate neuronal activity in V1: Effects of retinal image motion, position, and extraretinal influences. *Journal of Vision*, *8*(14). doi:Artn 1910.1167/8.14.19
- Leopold, D. A., & Logothetis, N. K. (1998). Microsaccades differentially modulate neural activity in the striate and extrastriate visual cortex. *Exp Brain Res*, *123*(3), 341-345. doi:10.1007/s002210050577
- Lowet, E., Gomes, B., Srinivasan, K., Zhou, H., Schafer, R. J., & Desimone, R. (2018). Enhanced Neural Processing by Covert Attention only during Microsaccades Directed toward the Attended Stimulus. *Neuron*, *99*(1), 207-214 e203. doi:10.1016/j.neuron.2018.05.041
- McFarland, J. M., Bondy, A. G., Saunders, R. C., Cumming, B. G., & Butts, D. A. (2015). Saccadic modulation of stimulus processing in primary visual cortex. *Nature Communications*, *6*. doi:ARTN 8110
10.1038/ncomms9110
- Meirovithz, E., Ayzenshtat, I., Werner-Reiss, U., Shamir, I., & Slovin, H. (2012). Spatiotemporal effects of microsaccades on population activity in the visual cortex of monkeys during fixation. *Cereb Cortex*, *22*(2), 294-307. doi:10.1093/cercor/bhr102
- Rezak, M., & Benevento, L. A. (1979). A comparison of the organization of the projections of the dorsal lateral geniculate nucleus, the inferior pulvinar and adjacent lateral pulvinar to primary visual cortex (area 17) in the macaque monkey. *Brain Res*, *167*(1), 19-40. doi:10.1016/0006-8993(79)90260-9
- Rockland, K. S., & Pandya, D. N. (1979). Laminar origins and terminations of cortical connections of the occipital lobe in the rhesus monkey. *Brain Res*, *179*(1), 3-20. doi:10.1016/0006-8993(79)90485-2

- Tse, P. U., Baumgartner, F. J., & Greenlee, M. W. (2010). Event-related functional MRI of cortical activity evoked by microsaccades, small visually-guided saccades, and eyeblinks in human visual cortex. *Neuroimage*, *49*(1), 805-816. doi:10.1016/j.neuroimage.2009.07.052
- van Kerkoerle, T., Self, M. W., Dagnino, B., Gariel-Mathis, M. A., Poort, J., van der Togt, C., & Roelfsema, P. R. (2014). Alpha and gamma oscillations characterize feedback and feedforward processing in monkey visual cortex. *Proc Natl Acad Sci U S A*, *111*(40), 14332-14341. doi:10.1073/pnas.1402773111
- Vanni, S., Hokkanen, H., Werner, F., & Angelucci, A. (2020). Anatomy and Physiology of Macaque Visual Cortical Areas V1, V2, and V5/MT: Bases for Biologically Realistic Models. *Cereb Cortex*, *30*(6), 3483-3517. doi:10.1093/cercor/bhz322
- Xing, D., Yeh, C. I., Gordon, J., & Shapley, R. M. (2014). Cortical brightness adaptation when darkness and brightness produce different dynamical states in the visual cortex. *Proc Natl Acad Sci U S A*, *111*(3), 1210-1215. doi:10.1073/pnas.1314690111
- Zanos, T. P., Mineault, P. J., Nasiotis, K. T., Guitton, D., & Pack, C. C. (2015). A sensorimotor role for traveling waves in primate visual cortex. *Neuron*, *85*(3), 615-627. doi:10.1016/j.neuron.2014.12.043

REVIEWER COMMENTS

Reviewer #1 (Remarks to the Author):

The authors have fully addressed all my comments. The results of this paper reveal a novel interaction between eye movements and visual responses generated at early stages of cortical processing (areas V1 and V2). The data is of high quality and the experiments are well executed. I have no further comments.

Jose Manuel Alonso

Reviewer #2 (Remarks to the Author):

The authors of this manuscript have carefully addressed all my concerns with new data, analysis, control models and interpretations. I appreciate their effort. This is an interesting observation and worth further investigation. I have no further questions.

Dear reviewers,

Below are our point-by-point responses to your comments. For easy identification, our responses are written in the style of bold and the corresponding reviewer comments were marked in the style of italics and underline in the file. The changes in the manuscript were highlighted in red color. Your comments were very encouraging and helpful, and we sincerely thank both of you for the constructive feedback. We have made substantial changes for figures, text, figure legends and supplementary information. Addressing the comments has improved and clarified the communication of the findings we report in our manuscript. We hope that our responses and revision are satisfactory to both of you.

Sincerely,
Dajun Xing

Reviewer #1 (Remarks to the Author)

This interesting paper demonstrates a novel neuronal mechanism underlying the effects of microsaccades on visual perception. It describes a series of carefully-designed and well-executed experiments that will have a major impact in our understanding of the relation between eye movements and vision. The paper first demonstrates that two main areas at early stages of the visual cortical processing (V1 and V2) are differently modulated by small eye movements called microsaccades, which span less than 1 degree in amplitude. They convincingly show that microsaccades modulate the activity of area V2 but not area V1 and that the modulation in area V2 is tuned to microsaccade direction and is strongest when the microsaccades move the eye away from the V2 neuronal receptive field. The authors provide two excellent controls to demonstrate that these V2 activity modulations are not driven by visual stimuli. They then show evidence that the microsaccade response modulation originates at the input layer of V2, is caused by response suppression, and occurs in monkeys that also have their visual perception affected by microsaccade direction. The paper also provides a computational model that reproduces the effect of microsaccades on V2 activity and visual behavior. I have no major criticisms or concerns with the results and main conclusions of the paper. I just have a few suggestions that I hope will help the authors to improve what is already a strong paper.

Reply: We appreciate your positive comments and recognition for our study. Thank you very much!

1. The controls that the authors show in Figure 2 are very convincing. *However, the evidence that microsaccade amplitude is not correlated with firing rate could be stronger.* For example, it would be helpful to show the correlation as a scatter plot and report the correlation value. This may be important to help understand the origin of the signals driving the microsaccade modulation of V2 activity. The two controls from Figure 2 are very effective at demonstrating that the V2 activity modulations are not driven by visual stimuli. Therefore, the authors are leading their readers to believe that the modulations are driven by attention and/or motor signals. In the discussion, the authors argue that attention is not the main driver of these modulations. Therefore, motor signals

appear to be the most likely source of these modulations. *However, if the V2 response modulations are driven by motor signals, I would expect to see some correlation between saccade amplitude and firing rate, even if it is weak* (perhaps, I would consider including in this analysis not just microsaccades but also saccades with amplitude > 1 deg). This is not a major criticism. The authors may have other reasons to show the control in the way they are showing it and they can decide what is the best approach to make their paper stronger.

Reply: Thank you for the nice and important suggestion, which helps us further understand neural origin for the directional modulation of microsaccades (MS). Following your suggestion, we did further analysis on the relationship between MS amplitudes and firing rates modulated by MS. In the old version of our manuscript, we only looked at the correlation between MS amplitudes and MS modulation without taking care of directions of these microsaccades. In the new correlation analysis, we separated microsaccades toward and away from RFs to investigate the possible interaction between MS direction and amplitude on modulating firing rates, and we provided result of V1 as a control.

We first showed the scatter plots of four example sites (Figure R1 a-b) where each dot represents a single microsaccade and its modulation on neural response. In these examples, only MSs away from RFs brought significant correlations between MS amplitude and firing rates in V2. However, due to large fluctuations of neural response during each MS, the linear trend was hard to detect and only a few sites exhibited significant correlation. We are unable to include large saccades (larger than 1 degree) in the analysis, because we have restricted the circular fixation window with a radius of 1 degree, and our data collection system discarded a trial if there was any microsaccade larger than 1 degree.

In order to obtain reliable modulation signals, we binned neural responses of each site (in V1 and V2) by amplitudes of all available microsaccades in ‘toward’ and ‘away’ directions respectively, and calculated correlation between mean amplitudes of binned microsaccades and their mean modulations for all sites (Figure R1 c-d). Interestingly, during the late period (126 ~ 450 ms, the period where we found direction modulation in V2), we found a significant correlation for ‘away’ direction but not for ‘toward’ direction (Figure R1d). The difference between the two correlation coefficients was significant ($z = 2.36, p = 0.02$). Such a weak but significant relationship was only observed in V2, but not in V1 (Figure R1c). The results held consistent when we did correlation analysis on the group means of modulation and ten levels of amplitudes (Figure R1e-f). The correlation in ‘away’ direction at late period supports the view that directional suppressive modulation after microsaccades in V2 might be related to motor signals.

We have added the correlation results in main text (in lines 203-214 on pages 10-11) and added the scatter plots (Figure R1e-f) as part of the new figure 3 (Fig. 3d) in the revision, as the evidence supporting the motor source for the direction-modulation of MS in V2. Thank you for the suggestion!

Figure R1 Relationship between microsaccade modulation and amplitude

a Correlation between microsaccade amplitude and averaged neural response during late period (126 ~ 450 ms) from two example sites in V1. **b** Correlation between microsaccade amplitude and averaged neural response during same period from two example sites in V2. **c** Normalized responses of each site (in V1) binned by amplitudes of microsaccades in 'toward' and 'away' directions respectively. Blue and orange dots indicate the mean responses in each amplitude bin. **d** Normalized responses of each site (in V2) binned by amplitudes of microsaccades in 'toward' and 'away' directions respectively. Solid blue line indicates the fitted line. **e** Correlation between averaged V1 response across units and microsaccade amplitude. Solid lines indicate fitted results. **f** Correlation between averaged V2 response across units and microsaccade amplitude. Confidence interval is shown as blue shadow.

2. The authors may want to leave open the possibility that the microsaccade modulation of V2 activity may be related to attention. *Attention not only enhances but also suppresses cortical responses. Moreover, even in the absence of a stimulus, the monkey may have a bias towards attending a spatial position where the stimulus has been previously shown multiple times.* There is not much to be gained by claiming that the V2 response modulation is not related with attention.

The attention effects related to microsaccades may be weak and easy to miss. If it is not an attention effect, the reader is forced to conclude that the effect is driven by some type of motor signal. If it is a motor signal, I would expect to see a correlation between the response modulation and saccade amplitude.

Reply: Thank you for the suggestion. We appreciate the solid logic behind your suggestion. Our response to your first comment suggests that the directional modulation of MS is very likely to come from motor source. However, we do agree with you about that attention not only enhances but also suppresses neural responses to irrelevant information. And we cannot fully exclude the attention effect that is time-locked to microsaccade since we did not manipulate attention itself in the fixation task. Following your suggestion, we have revised the discussion part (in line 437-453 on page 20) to leave open the possibility that the microsaccade modulation of V2 activity may be also related to attention; and we also raised the possibility that we might miss some weak attention effects related to microsaccades.

Other minor comments

1. Line 132. 'We found that the averaged V2 responses after microsaccades moved in different directions are well-tuned'. The term 'well-tuned' is unclear here because the term 'direction tuning' was not introduced yet in the text. It may better to write that '... the averaged cortical responses were tuned to microsaccade direction in area V2 (...) but not in area V1.'

Reply: We have changed the phrase 'well-tuned' to the descriptive sentence '... the averaged visual cortical responses were tuned to microsaccade direction in area V2 (...) but not area V1'. Please see the changes labeled in red color in lines 97-98 on page 6. Thank you for the suggestion!

2. Typo in line 201 (and other parts of this text section): 'faciliatory'. Consider changing to '...might be due to facilitation'.

Reply: We have changed 'faciliatory modulation' to 'facilitation' in main text and figure legends (please see changes in line 174 on page 9, line 460 on page 21 and in line 885 on page 35).

3. Figure 4B. Please, provide a color scale bar or define yellow and blue in the figure legend. The reader needs this information to understand the color code for response facilitation and suppression.

Reply: We apologize for our negligence. A color scale bar has been added in Figure 4B.

4. Figure 4D. The dots with different shades of gray are difficult to see. Consider using different colors instead (e.g. green, brown and black). What are the vertical lines superimposed on the asterisks? They are not explained in the figure legend.

Reply: Thank you for the suggestion. We have changed the color of dots to red, blue and yellow to represent sites from three monkeys. The vertical lines with asterisks denote significant result from post-hoc comparisons of Kruskal-Wallis test (Bonferroni-corrected). We have changed the location of asterisks in Figure 4d, and added the corresponding description in the figure legend (in lines 915 on page 36).

5. Figure 4C. The differences in latency are not easy to see in this figure panel (they are very clear in Figure 4D panel). Consider showing dotted lines and latency values next to each panel. Consider also adding labels to directly show in the figure what measurements are done in supragranular, granular and infragranular layers.

Reply: We defined latency by the first bin of time where there is a significant difference of responses between toward and away conditions in paired t test (Bonferroni-corrected). The time point with significant difference in the two directions has been shown as gray bars in Figure 4c, and we have added vertical lines in each panel to show the latency in each layer. The latency values were added in the figure legend (please see line 908-909 on page 36).

6. Figure 5B. It would be helpful to report in the figure panel or figure legend the luminance of the target and background in cd/m^2 . The units of target luminance in the x axis are not reported (is this luminance or luminance contrast?).

Reply: Thank you for the suggestion. The x axis now is labelled as contrast (defined as $(\text{luminance}_{\text{(target)}} - \text{luminance}_{\text{(background)}}) / \text{luminance}_{\text{(background}})$). We added the physical luminance value in cd/m^2 in the figure legend (please see lines 926-927 on page 36).

7. Line 336. Consider writing ‘... the target that appeared in...’.

Reply: Thank you for the suggestion. We have changed ‘... the target appeared in ...’ to ‘... the target that appeared in ...’ in line 289 on page 15.

8. Figure 6A. Define M and S input in figure panel or figure legend.

Reply: We apologize for the unclear statement of the model elements. To help readers’ understanding, we replaced S with ‘V1 biphasic response’ which took input of onset times of microsaccades. We have added the description of the model components (M and other components) in the figure legend (please see lines 937-943 on page 37).

9. Line 372. Typo. ‘V’ should be ‘V2’.

Reply: We would like to clarify that ‘V’ here represent the component V in the model stage 2, since the panel shows simulation results of mean-field activities after averaging V_L and V_R according to microsaccade direction. We have changed ‘V’ to ‘V in stage 2’ in line 950 on page 37 and changed the title of the panel (Figure 4e) to ‘stage 2’ to avoid misinterpretation. We also changed ‘V’ to ‘the stage 2 (V)’ in the other places in the text for the same reason. Thank

you!

10. Line 556. Did you use U probes or V probes? Methods reports V probes but the main text reports U probes.

Reply: We apologize for the mistake. We used V probes and we have changed 'U' to 'V' in line 61 on page 4.

Reviewer #2 (Remarks to the Author):

Microsaccades (MS) are a form of involuntary saccades which are smaller than 1 degree. MS have been shown to have effects on refreshing visual information. However, how MS modulates visual cortex is unclear and limited studies have shown direction specific modulation effect of MS in early visual cortex. This study investigated the MS influence on V1 and V2 neurons in macaque. Surprisingly, they found direction-specific suppression of MS in V2, but not in V1, and appeared first in the middle layer. They further tested the behavioral consequence by XX task and built a hierarchical recurrent model to link neural activity with behavior.

Major comments:

Why would neurons in V2 show significant modulation from MS, while V1 neurons showed no effect, regardless of the strong recurrence between the two areas? It would be helpful to rule out the following possibilities.

Reply: Thank you for the constructive suggestions, which help us rule out several possible V1 sources for the directional microsaccade (MS) modulation in V2. We have included more experimental data and performed further analyses to address each possibility you suggested. Results in responding to both your questions below and the question from the reviewer #1 (Figure R1) suggest that the directional modulation of microsaccades in V2 is more likely to come from motor system, and it is unlikely to directly come from V1.

You have also indicated a very interesting question, that is since there is a feedback loop (recurrence) from V2 to V1, why directional modulation by MS in V2 doesn't feedback to V1? We think this is mainly due to the following reasons. Anatomical studies (Rockland & Pandya, 1979; Vanni, Hokkanen, Werner, & Angelucci, 2020) have shown that the feedback loop from V2 to V1 is laminar specific, mainly between the deep layer of V2 and the superficial layer of V1 (Figure R2a). Based on the laminar pattern of directional modulation in V2 shown in Figure 4d (we also copy this figure as Figure R2b), the directional modulation of MS in the deep layer of V2 was significantly weaker than that in V2 middle layer. Therefore, V2 feedback (from V2 deep layer) effect on V1 might not have strong direction modulation of MS. This might explain why our results don't show significant direction modulation of MS in V1.

Figure R2 Recurrence between V1 and V2 and laminar profile of direction modulation strength

a Schematics of laminar circuits between V1 and V2. **b** Laminar distribution of direction modulation index in V1 and V2. The vertical black and cyan lines indicate the mean of each depth bin for each cortical region. Shaded regions denote the SE across units.

It is worth pointing out that, we did not see significant direction modulation of MS in V1, but we did see strong and significant non-direction modulation of MS in V1 (biphasic response in figure 1d), which is consistent with existing literatures (Hass & Horwitz, 2011; Kagan, Gur, & Snodderly, 2008; Meirovithz, Ayzenshtat, Werner-Reiss, Shamir, & Slovin, 2012). And the laminar pattern of biphasic modulation in V1 (figure 4b) was highly consistent with a previous study (McFarland, Bondy, Saunders, Cumming, & Butts, 2015): the suppression first appeared in the granular and infra-granular layers of V1, specifically layers 4 and 6. Compared to the strong non-directional biphasic modulation in V1, which indicates good V1 response, the directional effect in V1 was very weak. We speculate that strong non-directional modulation of MS might mask weak directional modulation of MS in V1 (due to feedback from V2). In the superficial layer of V1, we do identify a very weak trend of direction modulation of MS (L2/3: $t_{101} = 1.95$, $p = 0.054$, two-sided t test), which might indicate some recurrence from V2 to V1, although the direction modulation of MS is not significant in any cortical layers.

Overall, we think V1 might not be a major target of directional modulation by MS. Our results suggest that directional modulation is mainly from a motor source projecting to V2 middle layer and have a strong impact on downstream visual processing while show very weak effect in V1 via V2-V1 feedback modulation. Thank you for this comment, and we have added more discussion about this point in main text (please see lines 388-402 on page 18-19).

To check whether V1 have significant directional modulation of MS, we have carefully inspected all possibilities you suggested, by performing further analyses and including more experimental data elaborated below.

1. Could it be due to the RF location (central vs peripheral) of the recorded V1 and V2 neurons? I would like to see the distribution of center of RF for V1 and V2 units relative to fixation.

Reply: Thank you for the suggestion. We have added the distribution of RF centers in Figure 1b (main text: lines 64-65 on page 4) and explained the result here in more details. We estimated/mapped the receptive field of each unit in our study with small black/white squares rapidly (20 ms) and randomly presented at different location on the screen (Sparse Noise Experiment, Figure R3a, see method for details, lines 579-586 on page 26). We then used a two-dimensional gaussian function here to estimate the center and spatial profile of receptive fields (Figure R3b and c).

Figure R3 The distribution of RF for V1 and V2

- The trial procedure in the sparse noise experiment. Red dashed circle indicates the RF location of a recorded unit. The red dot represents the fixation point.
- RF fitting for one example unit in V1 and V2. Red circles indicate the RF with diameter of 4σ of fitted 2D gaussian.
- The spatial distribution of center of RFs for V1 and V2 units.
- The distribution of RF eccentricity for V1 and V2 units.

The spatial distribution for centers of RFs for V1 and V2 units, relative to fixation point (0,0) was shown in Figure R3c. The RFs of most units from V1 and V2 overlapped with each other and located within the eccentricity of 5° . The median of RF eccentricity of V1 units was 3.05° and the median of RF eccentricity of V2 units was 3.67° . And the difference of the two eccentricity medians (0.62 deg) was smaller than the size (diameter) of RFs of V2 neuron ($0.8\sim 1.2 \text{ deg}$). In our responses to your 3rd comment, we will further demonstrate that the eccentricities of V1 and V2 sites in our study have a negligible effect on the results (Figure R7).

2. Alternatively, is it because the stimulus size is 4-6 degree, which is much larger than the V1 receptive field size and induced a strong suppression in V1 to begin with? To rule out this factor, it would be helpful to use smaller stimulus size for a control.

Reply: Thank you for the nice suggestion of a control experiment with small size. We set stimulus size at $4^\circ\sim 6^\circ$ in the experiments, to guaranteed that RFs of V1 and V2 were fully covered by the surface of visual stimuli, and RFs would not be directly driven by the edge of a stimulus even with the consideration of RF shifts caused by microsaccades ($<1^\circ$). Such an experimental design made the response dynamic after microsaccade purely reflect a motor signal, without changing the visual stimulus within RFs of V1 and V2. To better rule out the possibility that surround suppression might reduce the directional modulation of MS in V1, as you recommended, we used some smaller sizes in a control experiment on monkey DQ and

DS. We recorded V1 neurons on DQ by a Utah array (Figure R4a) with a uniform disk shown in the center of RFs for 2 seconds. We found that surround suppression can change the amplitudes of non-directional modulations by MS; but there was no significant directional modulation of MS in V1, even when smaller sizes of stimulus was used ($<0.1^\circ$, $0.2\sim0.5^\circ$, 2.3° ; $p > 0.05$, paired *t* test with corrections for multiple comparisons). With data recorded in the V1 of monkey DS by a linear array (V-Probe), we still did not find significant directional modulation with smaller stimuli ($<0.1^\circ$, $0.3\sim0.6^\circ$) (Figure R4f-g). Based on the results of the control experiment on two monkeys, we hope you are convinced that the possibility that surround suppression in V1 impairs the potential directional modulation after microsaccade can be ruled out. We have added the results of the control experiment in supplementary information (Supplementary Fig. 9) and also mentioned in the discussion (please see lines 396-398 on page 19).

Figure R4 The control experiment with small size in V1.

- V1 recording with a Utah array in monkey DQ.
- Normalized MUA in V1 with a small circle ($<1^\circ$) represent on the center of RFs (orange and blue lines in the left vertical axis) and averaged velocity of eye movement (the black line in the right vertical axis) around microsaccade onset.
- Same with b but the stimulus size was $0.2\sim0.5^\circ$.
- Same with b and c but the stimulus size was 2.3° .
- Laminar recording in V1 in monkey DS.
- Same with b in monkey DS.
- Same with f but the stimulus size was $0.3\sim0.6^\circ$

We provided another way to test whether the difference of V1 and V2 on direction-specific microsaccade modulation is due to response excitation/suppression (caused by factors such as surround suppression). We checked whether direction-specific microsaccade modulation is different for recording sites at different response levels, by analyzing the relationship between

direction modulation index and baseline firing rates before microsaccades. We did not find any significant correlation in either V1 ($r = 0.07$, $p = 0.199$) or V2 ($r = 0.11$, $p = 0.051$). We further divided all channels into 5 groups based on quintiles of their baseline firing rates before microsaccades (Figure R5), and we did not find significant directional modulation of microsaccades in any level of baseline firing rates in V1. In V2, all groups exhibited significant directional modulation. This result indicates that the level of excitation/suppression during microsaccade generation in visual cortex might not influence the directional modulation strength. We have added the results of control for baseline firing rates in Figure 2b and in the main text (please see lines 149-156 on page 7-8). Thank you for the advice!

Figure R5 Correlation between baseline firing rates and direction modulation index
a Distribution of normalized baseline firing rates before microsaccades of V1 and V2 channels. **b** Direction modulation index of V1 and V2 populations in each baseline firing rates groups defined by five quintiles.

3. Based on the experimental design, both V1 and V2 units are simultaneously recorded. However, there are inconsistency of sessions been analyzed in each monkey between V1 and V2. Would this cause a difference in the observed V1 and V2 difference?

Reply: We apologize for not describing the experimental design clearly. The 24-channel probe (~2.66 mm) was not long enough to simultaneously record full layers in both V1 (~2 mm) and V2 (~1.3mm). We have to record one region at a time to guarantee the whole laminar was covered. The recording sessions for V1 and V2 were alternated in different days due to the limited trial number that monkeys could bear in one day. That's why the sessions numbers in V1 and V2 for each monkey is not the same. We have added a brief description of the recording method in the main text (in lines 62-64 on page 4).

Among the 32 sessions of V2 recordings, there were 18 sessions that all layers of V2 and part of the above V1 (middle and deep layers) were simultaneously recorded by a single probe.

One example penetration and the relative RF locations of V1 and V2 units were shown in Figure R6a. In these sessions, the stimulus was large enough to cover RFs of all recorded units (in both V1 and V2). These V1 units ($n = 104$) were not used in the old version of manuscript but we included them into the analysis for simultaneous recording in V1 and V2 as a control for the session number. We found the modulation strength in V1 after microsaccade with two directions was comparable, while in V2 firing rates after microsaccades towards RFs were significantly higher than firing rates after microsaccades away from RFs (Figure R6b). And the population distribution of directional modulation index (DMI) for V2 units was higher than DMI distribution for V1 units (Figure R6c). The results of simultaneous-recording sessions with a single probe were consistent with our main findings from separate recordings in V1 and V2 shown in Figure 1. We have replicated the findings in Figure 1, 2 and 3 by using the simultaneous-recording data, which were shown in the supplementary information (Supplementary Fig. 5) and were mentioned in the main text (in lines 117-124 on pages 6-7, lines 147-149 on page 7, lines 194-202 on page 10). Thank you for the question!

Figure R6 Simultaneous recordings in V1 and V2

- Simultaneous recording in V1 and V2 and the RF mapping for each channel from an example session.
- Normalized MUA in V1 and V2 with a square represent on the center of RFs of V1 and V2 units (orange and blue lines in the left vertical axis) and averaged velocity of eye movement (the black line in the right vertical axis) around microsaccade onset. Gray bar on the top indicates the significant time period in the *paired t*-test with Bonferroni corrections.
- Population distribution of direction modulation index of each channel in V1 and V2. '****' indicates $p < 10^{-10}$ and 'ns' indicates $p > 0.05$ in two-sided *t*-test.

Your question also enlightened us to provide another control analysis for comparing directional modulation by MS in V1 and V2 with overlapping RFs but at different sessions of probe recordings from the same monkeys. Usually overlapping RFs in V1 and V2 indicate the recording column in V2 receive direct feedforward projections mainly from the corresponding

column in V1, which can help to dissect the input from V1 and new processing mechanism in V2 without the potential confounding of column diversity. A potential drawback for simultaneous recording by a single probe was that the RFs of V1 and V2 don't overlap due to the folding structure of cortex. Considering this point, we selected the sessions with overlapping RFs in V1 and V2 within each monkey (Figure R7a) and did the same analysis as Figure 1 to compensate the drawback of simultaneous recording of one probe. We found similar results that V2 exhibited directional modulation from time course of firing rates, direction tuning and population distribution while the modulation in V1 did not show significant difference between toward and away conditions (Figure R7). We have added the results from units with overlapping RFs in supplementary information (Supplementary Fig. 8) and also in the discussion (lines 394-396 on page 19).

Figure R7 microsaccade modulation in V1 and V2 with overlapping RFs

- RFs of V1 and V2 units were overlapping from 18 V1 sessions and 12 V2 sessions.
- Normalized MUA (orange and blue lines in the left vertical axis) in V1 (top) and V2 (down) and averaged velocity of eye movement (the black line in the right vertical axis) around microsaccade onset. Gray bar on the top indicates the significant time period in the *paired t*-test with Bonferroni corrections.
- Distribution of direction modulation index in V1 and V2.

We hope to further answer your question by using the simultaneous recording data (7 sessions) from another project/experiment on monkey DS (Figure R8a). In the new experiment, we use two probes to simultaneously record V1 and V2 units with overlapping RFs (Figure R8c). We again found similar results as we shown in figure 1, that is V1 did not show directional modulation by MS while V2 did (Figure 8d-e). Considering the two kinds of simultaneous recording with the same session number in V1 and V2, we hope you are satisfied with our answers.

Figure R8 Simultaneous laminar recordings with two probes in V1 and V2 in monkey DS

- Simultaneous laminar recording in V1 and V2 with two linear probes.
- Current source density pattern in V1 and V2 after stimulus onset.
- The overlapping RFs of 11 units from V1 and 11 units from V2 in one example session.
- Normalized MUA in V1 and V2 (orange, MS toward; blue, MS away) and averaged velocity of eye movement (the black line in the right vertical axis) around micro-saccade onset. There were 7 sessions in total. Gray bar on the top indicates the significant time period in the *paired t*-test with corrections for multiple comparisons.
- Population distribution of direction modulation index of each channel in V1 and V2. **** indicates $p < 10^{-10}$ and 'ns' indicates $p > 0.05$ in two-sided *t*-test.

4. Is there any animal specific MS modulation effect? I would like to see a supplemental figure plotting the effect of MS on V1 and V2 units in the three macaque monkeys separately. I noticed there are significantly more sessions in DQ with V2 recording than V1 recording, and also more than number of sessions in other monkeys. I wonder whether this will introduce any bias.

Reply: Thank you for the suggestion. Results from the three monkeys separately are shown in Figure R9 and we have added the figure into supplemental materials (Supplementary Fig. 2) and added the description in the main text (in lines 92-94 on page 6). Results from each individual monkey are consistent with what we reported by combining data from all three monkeys. As Figure R9 shows, firing rates in V1 after microsaccades in the toward and away conditions showed comparable strength of modulation ($p > 0.05$, *paired t*-test with corrections) for all three monkeys, while firing rates after microsaccades toward RFs were higher than those away from RFs ($p < 0.05$, *paired t* test with corrections for multiple comparisons) in V2 for all three monkeys.

To further test whether the session number of V2 recording on monkey DQ (17 sessions) brought any bias when we comparing V1 and V2 on monkey DQ, we randomly select 5 sessions from 17 sessions and showed the MS effect in Figure R9d. We found even when the number of channels (N = 48) in V2 selected sessions was less than the number in V1 (N = 52, 6 sessions),

the directional modulation of microsaccades was still significant ($p < 0.05$, *paired t test* with corrections). Next, to test the result consistency of subset of V2 sessions from DQ and whether the number of V2 recording sessions from DQ brought any bias on direction modulation effect in V2, we use bootstrapping method to compare direction modulation strength of DQ to two other monkeys. We randomly sampled 8 sessions from 17 sessions (comparable to 9 V2 sessions in DS and 7 V2 sessions in DK) and calculated the average direction modulation index of the subset of data. This procedure was repeated for 200 times and the distribution of means of direction modulation index was shown in Figure R9e. All means of subsets from DQ V2 data were above zero, indicating a consistent and robust directional modulation in V2 of DQ. Based on the percentile test, we did not find any significant difference between the direction modulation strength (DMI) of subsets of DQ V2 sessions and modulation strength of V2 population from DS ($p > 0.05$) or any difference between DMI of subsets of DQ and DMI of V2 population from DK ($p > 0.05$). Furthermore, all three monkeys exhibited significant direction modulation of V2 in a population level ($p < 0.001$, one-sample *t test*).

Based on the above analysis, data from monkey DQ DK and DS all showed significant and comparable strength of direction-specific microsaccade modulation in V2, but only non-directional modulation by MS can be found in V1. This suggests that different numbers of sessions from the three monkeys were unlikely to cause a bias for estimating the direction modulation in V1.

Figure R9 Microsaccade modulation in three monkeys

- a. Normalized MUA (orange and blue lines in the left vertical axis) in V1 (top) and V2 (down) from monkey DQ and averaged velocity of eye movement (the black line in the right vertical axis) around microsaccade onset. Gray bar on the top indicates the significant time period in the *paired t-test*.

- b. Normalized MUA in V1 and V2 from monkey DK and averaged velocity of eye movement around microsaccade onset.
- c. Normalized MUA in V1 and V2 from monkey DS and averaged velocity of eye movement around microsaccade onset.
- d. Normalized MUA and velocity of eye movement from 5 randomly selected session of monkey V2 recordings.
- e. The distribution of means of direction modulation index (DMI) for bootstrapped sessions. Vertical solid lines represent the means of DMI for all sessions from DS (yellow) DK (blue) and DQ (red). Asterisks donates statistical significance ($p < 0.001$).

5. I also have some doubt about the V1 data quality. Based on the relative power plot of V1 recording sites shown in supplement figure 2, this is no obvious gamma peak given the size of the stimulus (like the peak around 40Hz in V2). I wonder maybe the reason this study didn't observe effect in V1 is due to poor data quality in V1.

Reply: The broad gamma peak in V1 (in Supplementary Fig.6) is due to short time window (150–450 ms after microsaccades) available for us to conduct power spectrum analysis on the LFP after a microsaccade, which is much narrower than that used in traditional spectrum analysis of the LFP (usually longer than 1s). When we use longer time window (0.5s-3s after stimulus onset) to perform spectrum analysis on the LFP from the same dataset, we can get a much clearer gamma peak in V1 (Figure R10), which is consistent with previous studies (Xing, Yeh, Gordon, & Shapley, 2014). To further test whether the absent of direction-specific microsaccade effect in V1 is due to poor quality of the V1 data in some channels, we directly inspected the relationship between directional modulation of MS and data quality of spike activity from recording sites, defined as signal-noise ratio (SNR), in V1 (Figure R11).

Figure R10 LFP power spectrum for different surface luminance in V1 and V2

- a. LFP power spectrum in V1 after stimulus onset (0.5 – 3s).
- b. LFP power spectrum in V2 after stimulus onset (0.5 – 3s).

The signal-noise ratio (SNR) of spike activity for each recording site was defined by the site's relative visual response and its baseline activities:

$$SNR = \frac{\text{mean}(MUA_{\text{initial visual response in } 40-90\text{ms}}) - \text{mean}(\text{baseline MUA})}{\text{std}(\text{baseline MUA})}$$

We defined a channel as a good channel, if the site's SNR was higher than 0.65; in this way we got 278 good channels in V1 and 283 in V2 as shown in Figure R11a. Then we did the same analysis as that for Figure 1 and the results were similar to our findings with the whole dataset: in V1, no significant difference on firing rates after microsaccade toward and away from RFs was found in time course ($p > 0.05$, *paired t-test* with multiple corrections) or spatial tuning; but in V2, significant directional modulation of firing rates after microsaccades and bell-shaped microsaccade direction tuning were found (Figure R11b). When we change the criteria for defining good channels (from SNRs >0.65 to SNRs > 2.5), the above conclusions can still be held (Figure R11d-f). The mean of direction modulation index of the neural population in V1 is again not significant different from zero.

Figure R11 Microsaccade modulation on MUA in V1 and V2 with good channels

- Normalized firing rates around microsaccade in V1 and V2 after selecting channels with SNR > 0.65 .
- Microsaccade direction tuning for the averaged responses of V1 and V2 from the period 125–450 after microsaccades, red and blue shadow regions indicate two direction bins used for calculated direction modulation index in **c**.
- Distribution of direction modulation strength in V1 and V2.
- Same as **a** but the units were selected by the criteria of 2.5.
- Same as **b** for units with SNR over 2.5.
- Same as **c** for units with SNR over 2.5.

We also analyzed the LFP of the channels with good quality as we did in Figure R11 to confirm the findings about gamma (Figure R12). In V1, we found no significant difference between gamma power (50-85 Hz) after microsaccades towards and away from RFs and we did not find a significant higher-than-zero distribution of direction modulation index of gamma (Figure R10b, 0.02 ± 0.30 , $M \pm SD$, $t(277) = 1.14$, $p = 0.26$). In V2, the gamma-band power was higher after microsaccades moved in the toward direction than after microsaccades moved in the away direction (0.20 ± 0.26 , $M \pm SD$, $t(282) = 12.56$, $p < 0.001$). When we change the criteria for defining good channels (from SNRs >0.65 to SNRs > 2.5), the above conclusions can still be held (Figure R12c-d). Both MUA and LFP results hold consistent after we chose the good channels.

Figure 12 Microsaccade modulation on LFP in V1 and V2 with good channels

- Relative power spectrum (to blank) of LFP in V1 and V2 with units of SNR over 0.65.
- Distribution of direction modulation strength of gamma power in V1 and V2. '****' indicates $p < 10^{-5}$ and 'ns' indicates $p > 0.05$ in two-sided t -test.
- Same as a but for units with SNR over 2.5.
- Same as b but for units with SNR over 2.5.

We further divided our dataset into 5 bins based on quantiles of signal noise ratio (Figure R13), in order to check the relationship between signal quality and direction modulation. We found in V2, groups in all bins at different level of SNRs exhibited significant direction modulation of microsaccades ($ps < 0.001$, after corrections for multiple comparisons) while in V1 we again did not find any significant direction modulation of microsaccade even in the bin with highest SNRs ($ps > 0.05$, after corrections for multiple comparisons). We also tried SNR calculated based on the sustained response in a longer time window where microsaccades were selected (500-2600ms after stimulus onset) and the results were similar and consistent (V2: $ps < 0.001$; V1: $ps > 0.05$). The correlation between direction modulation index and SNR also doesn't show any significance either in V1 or V2 (V1: $r = -0.00$, $p = 0.99$, $N = 358$; V2: $r = 0.09$, $p = 0.10$, $N = 329$). These results indicate that the data quality evaluated by visual response has little influence on the directional modulation of microsaccade. Considering the consistent directional modulation in V2 across three monkeys, we think the signal quality of current data might not introduce bias into results for V1. We have reported results for the control of signal noise ratio in the main text (please see lines 112-116 on page 6) and added them in the supplementary information (Supplementary Fig. 4-6). Thank you for the advice!

Figure R13 Direction modulation index of V1 and V2 was consistent in different levels of signal noise ratio
a Distribution of signal noise ratio for all units in V1 and V2. **b** Direction modulation index of V1 and V2 populations in SNR groups defined by quintiles. *** indicates $p < 0.001$ and **** indicates $p < 10^{-5}$ in two-sided t -test.

It will make the conclusion more solid, if these concerns can be resolved.

Reply: Thank you very much for raising the above concerns which really help us check our data carefully and provide more solid evidences to support our findings. We hope our responses are satisfactory to you.

The hierarchical recurrent model is a nice integration of both experimental and behavioral results. It would be helpful if the motivation for model design and assumptions can be clearly stated. What component is necessary for this model to realized its prediction, especially the dynamics? How about alternative/control models?

Reply: Thank you for the wonderful comment and suggestion. We have added more texts to clearly state the motivations for model design and assumptions for model simulation in main text (lines 296-303 on pages 15). Briefly, we'd like to express the idea that given that V2 units exhibited significant directional microsaccade modulation in the middle layer which receives V1 feedforward inputs, we hypothesized that a spatial-specific projection targeting in V2 could lead to biased visual sensitivity of monkeys' behavior in a visual sensitivity task. To test whether such selective input to V2 would be sufficient for interpreting behavioral dynamic after microsaccades, we constructed a hierarchical dynamic neural network which could make decision in a 2AFC task based on dynamic responses in V2. The model was composed of two feedforward stages (corresponding to V1 and V2) and a recurrent decision stage which was inspired by the simplified decision model that was well defined and widely used to explain the behavioral choice in reaction time tasks (Figure R14a).

We first validated our model structure (without any microsaccade modulation components) in the current detection task by replicating contrast-tuning curve measured psychophysically (Fig. 5b). When changing the magnitude of visual input, the accuracy curve of hierarchical dynamic model could be well fitted by a Naka-Rushton function (Fig. 14b) which shows consistent pattern with monkeys' behavior (Fig. 5b). The result indicates good validity of model structure on the 2AFC detection task. Then, we added microsaccade-related biphasic modulation signal to stage 1 and directional suppressive signal to stage 2 (simulating

V2) based on our experimental findings to investigate their contributions on the directional microsaccade effect in behavior dynamic (full model).

Your suggestion of a control model is a good point to contrast contributions of different components. We have added a control model by removing directional suppressive component *M* from the full model. In order to test whether the detection performance of the control model and the full model is also directionally impaired by microsaccade after microsaccades, we chose the input strength generating a correct rate at 75% in the task (Figure R14b) as what we did in a real monkey experiment, and then simulated all behavioral sessions in a trial-by-trial basis for 15 times. We averaged the simulation results and found that the performance of a control model exhibited decreased accuracy during microsaccade generation without any directional effect (Figure R14d). In contrast, when a directional specific suppression was introduced to the stage 2 of the model (the full model), the directional modulation emerged after microsaccades (Figure 14e). The comparison between the control model and full model indicates that the component *M* (directional suppression in V2) may contribute the directional modulation on behavior dynamics after microsaccades. We have added the results of the control models in Figure 6 and in the main text (lines 340-348 on page 17)

Figure R14 Performance of the full model and control model on predicting behavior dynamics

a Illustration for the structure of the dynamic recurrent network. The model is composed of two visual stages (*P* and *V*) that receive visual input, a mutual inhibited decision stage, and two modulation components (biphasic kernel and *M*) that take input of eye movements (including microsaccade time and direction). *P* was stimulated by a 20-ms impulse current when there is a visual target in the RF. Each microsaccade generation will trigger a biphasic modulation on both components in stage 1 and impulsive suppression in *M*. *M* only suppresses *V* with RF in the opposite direction of

microsaccades. **b** Mean accuracy (across 30 simulations, each simulation was composed of one repetition of all possible target location and target onset time) for each level of visual input strengths. Error bars represent standard deviation of accuracy across simulations. **c** Change in detection accuracy in monkey DN and DS as a function of target onset time relative to a microsaccade. The red vertical line indicates the onset time of microsaccades. The time courses were evaluated with bootstrap tests and hypothesis tests using the percentile method. Asterisks denote statistical significance ($p < 0.05$ after corrections for multiple comparisons). **d** Averaged simulated behavior accuracy of control model after removing component M across 15 simulations (all trials from all empirical sessions from one monkey compose one simulation). Shaded region represents the standard deviation across simulations. **e** Averaged simulated behavior accuracy of the full model across 15 simulations.

Some additional discussion points worth adding to the current study:

1. Computational importance to have direction-specific microsaccades modulation.

Reply: The suggestion is important for us to understand the functional meaning of our finding. In the revision, we have provided some further insights on the computational importance (lines 411-436 on pages 19-20), and we showed them here.

First, the presence of information about microsaccade direction in visual cortex may be important for visual stability which requires knowledge of eye movements. The brain needs to know the direction of the eye movement to avoid misinterpretation of moving environment during microsaccades (Cavanaugh, Berman, Joiner, & Wurtz, 2016). Previous anatomical studies have found that signals from oculomotor system to V2 relies on topographic projections (Adams, Hof, Gattass, Webster, & Ungerleider, 2000; Benevento & Rezak, 1976; Rezak & Benevento, 1979), which might be the neural basis for direction information in visual cortex. The directional suppression in V2 and in the downstream area could be used for computing and inferring whether the information updating in the brain was due to a rightward microsaccade or leftward move in visual space. The hypothesis was consistent with results in our control experiment of the moving surface (Figure 2d), which showed V2 neurons could reveal the difference between the object moving and the microsaccades.

Second, the directional modulation might help the visual cortex reset excitability. The continuous changes of suppression strength in cortical space of V2 after microsaccades result in unbalanced cortical excitability. For a subpopulation whose RFs are in the opposite direction of microsaccades, the feedforward visual stream from V1 needs to be stronger to counterbalance the suppression, while the visual stream to another subpopulation with RFs in the direction of microsaccades more easily reached the threshold. We found that gamma-band activity, which represent the processing of feedforward visual information in visual cortex (van Kerkoerle et al., 2014), was also modulated by microsaccade direction. The unbalanced strength when driving downstream cortical regions leads to a reset of spatial visual sensitivity after each microsaccade.

2. Reconciling with the inconsistent literature that didn't report direction-specific MS modulation.

Reply: Thank you for the suggestion. This is also an important point to discuss. V2 area of the monkey is hard for recordings, because most of V2 area is not on the surface of the brain

and they are underneath V1 area (see figure 1a for demonstration). Therefore, only three studies, to our best knowledge, have investigated microsaccade modulation in V2 (Leopold & Logothetis, 1998; Meirovithz et al., 2012; Tse, Baumgartner, & Greenlee, 2010). The neurophysiology study (Meirovithz et al., 2012) by voltage-sensitive dye imaging found that microsaccades induced monophasic response to small stimuli and induced biphasic response to large stimuli. The study of single cell recording (Leopold & Logothetis, 1998) also found strong monophasic enhancement in V2 following microsaccade when presenting a small grating in the RFs. And in the human functional MRI study, microsaccade induced an increase in BOLD signal peaking at 4.5 seconds then followed by a decrease below baseline with a large polar grating in the screen (Tse et al., 2010). However, none of them has focused on the question of how microsaccade modulates neural activity in V2 in a directional manner, which reflect the new perspective of current study. In addition to the lack of analysis on directional specific modulation by microsaccades in V2, the displacement of a pattern stimulus in RFs due to microsaccades also makes it difficult to determine the direction modulation from the extra-retina source, which is also the case for studies in the middle temporal (MT) region (Bair & O'Keefe, 1998; Herrington et al., 2009).

Our findings in V2 indicate the importance of microsaccade direction relative to RFs for determining its effect on neural activity, which is nonnegligible when trying to explaining the behavioral benefit of microsaccade toward a visual target. One study (Zanos, Mineault, Nasiotis, Guitton, & Pack, 2015) found the amplitude of travelling wave in V4 was larger after large saccade toward RFs, which is consistent with the current perspective in our study. And a more recent study of microsaccade found that only microsaccades toward RFs will follow attentional enhancement on firing rates of V4 neurons (Lowet et al., 2018). These two studies together with our current findings support a new view when taking microsaccade direction into consideration. The recent studies (Lowet et al., 2018; Zanos et al., 2015) on directional modulation of microsaccade in V4 area also indicate that there hasn't been any similar study in V2, the upstream area for V4.

We have added more texts to discuss our findings in V2 regarding to existing literatures (please see lines 371-387 on page 18, lines 403-410 on page 19). Thank you for the suggestion.

Minor comments:

1. line 65: "we report that direction-specific microsaccade modulation (DSMM) emerges in V2": it is better to use 'observed' instead of 'emerges'

Reply: We have changed the word 'emerges' to 'observed' in line 49 on page 3. Thank you for the suggestion.

2. The term 'V2 input layer' is confusing, since inputs to V2 can be from either bottom up or top-down. Please be more specific. Replace with middle layer.

Reply: we have changed the 'V2 input layer' to 'middle layer of V2' in line 18 on page 2 and line 216 on page 11.

3. "suppressive mechanism": the suppressive effect observed in the study belong to observation rather than mechanism

Reply: Thank you for the suggestion. we have changed the word ‘mechanism’ to ‘effect’ in line 50 on page 3 and changed ‘suppressive mechanism’ to ‘suppression’ in line 296 on page 15.

Reference

- Adams, M. M., Hof, P. R., Gattass, R., Webster, M. J., & Ungerleider, L. G. (2000). Visual cortical projections and chemoarchitecture of macaque monkey pulvinar. *J Comp Neurol*, *419*(3), 377-393. doi:10.1002/(sici)1096-9861(20000410)419:3<377::aid-cne9>3.0.co;2-e
- Benevento, L. A., & Rezak, M. (1976). The cortical projections of the inferior pulvinar and adjacent lateral pulvinar in the rhesus monkey (*Macaca mulatta*): an autoradiographic study. *Brain Res*, *108*(1), 1-24. doi:10.1016/0006-8993(76)90160-8
- Cavanaugh, J., Berman, R. A., Joiner, W. M., & Wurtz, R. H. (2016). Saccadic Corollary Discharge Underlies Stable Visual Perception. *J Neurosci*, *36*(1), 31-42. doi:10.1523/JNEUROSCI.2054-15.2016
- Hass, C. A., & Horwitz, G. D. (2011). Effects of microsaccades on contrast detection and V1 responses in macaques. *J Vis*, *11*(3), 1-17. doi:10.1167/11.3.3
- Kagan, I., Gur, M., & Snodderly, D. M. (2008). Saccades and drifts differentially modulate neuronal activity in V1: Effects of retinal image motion, position, and extraretinal influences. *Journal of Vision*, *8*(14). doi:Artn 1910.1167/8.14.19
- Leopold, D. A., & Logothetis, N. K. (1998). Microsaccades differentially modulate neural activity in the striate and extrastriate visual cortex. *Exp Brain Res*, *123*(3), 341-345. doi:10.1007/s002210050577
- Lowet, E., Gomes, B., Srinivasan, K., Zhou, H., Schafer, R. J., & Desimone, R. (2018). Enhanced Neural Processing by Covert Attention only during Microsaccades Directed toward the Attended Stimulus. *Neuron*, *99*(1), 207-214 e203. doi:10.1016/j.neuron.2018.05.041
- McFarland, J. M., Bondy, A. G., Saunders, R. C., Cumming, B. G., & Butts, D. A. (2015). Saccadic modulation of stimulus processing in primary visual cortex. *Nature Communications*, *6*. doi:ARTN 8110
10.1038/ncomms9110
- Meirovithz, E., Ayzenshtat, I., Werner-Reiss, U., Shamir, I., & Slovin, H. (2012). Spatiotemporal effects of microsaccades on population activity in the visual cortex of monkeys during fixation. *Cereb Cortex*, *22*(2), 294-307. doi:10.1093/cercor/bhr102
- Rezak, M., & Benevento, L. A. (1979). A comparison of the organization of the projections of the dorsal lateral geniculate nucleus, the inferior pulvinar and adjacent lateral pulvinar to primary visual cortex (area 17) in the macaque monkey. *Brain Res*, *167*(1), 19-40. doi:10.1016/0006-8993(79)90260-9
- Rockland, K. S., & Pandya, D. N. (1979). Laminar origins and terminations of cortical connections of the occipital lobe in the rhesus monkey. *Brain Res*, *179*(1), 3-20. doi:10.1016/0006-8993(79)90485-2

- Tse, P. U., Baumgartner, F. J., & Greenlee, M. W. (2010). Event-related functional MRI of cortical activity evoked by microsaccades, small visually-guided saccades, and eyeblinks in human visual cortex. *Neuroimage*, *49*(1), 805-816. doi:10.1016/j.neuroimage.2009.07.052
- van Kerkoerle, T., Self, M. W., Dagnino, B., Gariel-Mathis, M. A., Poort, J., van der Togt, C., & Roelfsema, P. R. (2014). Alpha and gamma oscillations characterize feedback and feedforward processing in monkey visual cortex. *Proc Natl Acad Sci U S A*, *111*(40), 14332-14341. doi:10.1073/pnas.1402773111
- Vanni, S., Hokkanen, H., Werner, F., & Angelucci, A. (2020). Anatomy and Physiology of Macaque Visual Cortical Areas V1, V2, and V5/MT: Bases for Biologically Realistic Models. *Cereb Cortex*, *30*(6), 3483-3517. doi:10.1093/cercor/bhz322
- Xing, D., Yeh, C. I., Gordon, J., & Shapley, R. M. (2014). Cortical brightness adaptation when darkness and brightness produce different dynamical states in the visual cortex. *Proc Natl Acad Sci U S A*, *111*(3), 1210-1215. doi:10.1073/pnas.1314690111
- Zanos, T. P., Mineault, P. J., Nasiotis, K. T., Guitton, D., & Pack, C. C. (2015). A sensorimotor role for traveling waves in primate visual cortex. *Neuron*, *85*(3), 615-627. doi:10.1016/j.neuron.2014.12.043

REVIEWERS' COMMENTS

Reviewer #1 (Remarks to the Author):

The authors have fully addressed all my comments. The results of this paper reveal a novel interaction between eye movements and visual responses generated at early stages of cortical processing (areas V1 and V2). The data is of high quality and the experiments are well executed. I have no further comments.

Jose Manuel Alonso

Reviewer #2 (Remarks to the Author):

The authors of this manuscript have carefully addressed all my concerns with new data, analysis, control models and interpretations. I appreciate their effort. This is an interesting observation and worth further investigation. I have no further questions.